# Chaos Moth Flame Algorithm for Multi-Objective Dynamic Economic Dispatch Integrating with Plug-In Electric Vehicles

Wenqiang Yang [1], Xinxin Zhu [1], Fuquan Nie [1,*], Hongwei Jiao [1], Qinge Xiao [2] and Zhile Yang [2]

1   School of Mechanical and Electrical Engineering, Henan Institute of Science and Technology,
    Xinxiang 453003, China; yangwqjsj@163.com (W.Y.); zxx2575438269@163.com (X.Z.);
    jiaohongwei@hist.edu.cn (H.J.)
2   Shenzhen Institutes of Advanced Technology, Chinese Academy of Sciences, Shenzhen 518055, China;
    qg.xiao@siat.ac.cn (Q.X.); zl.yang@siat.ac.cn (Z.Y.)
*   Correspondence: lyywlnyx@sina.com

**Abstract:** Dynamic economic dispatch (DED) plays an important role in the operation and control of power systems. The integration of DED with space and time makes it a complex and challenging problem in optimal decision making. By connecting plug-in electric vehicles (PEVs) to the grid (V2G), the fluctuations in the grid can be mitigated, and the benefits of balancing peaks and filling valleys can be realized. However, the complexity of DED has increased with the emergence of the penetration of plug-in electric vehicles. This paper proposes a model that takes into account the day-ahead, hourly-based scheduling of power systems and the impact of PEVs. To solve the model, an improved chaos moth flame optimization algorithm (CMFO) is introduced. This algorithm has a faster convergence rate and better global optimization capabilities due to the incorporation of chaotic mapping. The feasibility of the proposed CMFO is validated through numerical experiments on benchmark functions and various generation units of different sizes. The results demonstrate the superiority of CMFO compared with other commonly used swarm intelligence algorithms.

**Keywords:** chaos moth flame algorithm; dynamic economic dispatch; grid fluctuation; plug-in electric vehicles; global optimization; chaotic map

## 1. Introduction

In the stage of power system planning and operation, economic dispatch (ED) plays a critical role in maintaining financial benefits due to the high value of fossil fuels used in power plants. A minor improvement in ED may result in significant cost savings. Moreover, it is imperative to address the problem of power grid fluctuations, as fluctuations in power generation can cause significant damage to electrical equipment and result in substantial economic losses. However, in practice, precisely modeling ED is difficult as it encompasses various factors such as the valve point effects and transmission losses, making it non-convex, non-smooth, non-linear, and non-differentiable. Therefore, evolutionary computation techniques are usually chosen to solve such problems, including genetic algorithms (GA), particle swarm optimization (PSO), ant colony optimization (ACO), and others.

Especially in recent years, many meta-heuristic algorithms have been applied to ED problems, including many new and improved algorithms that have gained some achievements. For example, the cogeneration is introduced into an ED problem that can achieve energy saving and improve environmental quality, which is solved by the cuckoo search algorithm (CSA) with penalty function. However, only low-dimensional cases have been studied, and experiments and analyses of high-dimensional cases are lacking [1]. A new parallel hybrid meta-heuristic method, combining the hybrid topology binary particle swarm optimization algorithm, adaptive differential evolution algorithm, and lambda iterative method, is proposed to solve the ED problem; however, no smart demand side management is achieved [2]. A chaotic local search-based bee optimization algorithm is

utilized for the ED problem, which has the advantages of fast convergence and strong local search ability [3]. A short-term load ED problem combines the ED problem with machine learning-based short-term load forecasting (STLF), where a new energy dispatch model is proposed and solved by a new dynamic genetic algorithm; however, experiments were conducted on only three test systems and no other new energy generation methods were introduced [4]. Due to the uncertainty of wind power, the wind speed model is considered in the ED problem, which is conducive to exploring a more balanced low-carbon power dispatching strategy for wind power integrated systems, and is investigated by the flower pollination algorithm to solve this class of economic dispatching problems [5,6]. The chaotic map is utilized to improve the performance of the bat algorithm, which is presented to solve the static economic dispatch problem [7]. The cuckoo search algorithm (CSA) is applied to ED problems where wind turbines and fuel cells are considered, and a cuckoo search algorithm for microgrid power dispatch problems has also been proposed with some prospect [8]. The fuzzy dominance is used to select Pareto optimal front (POF) and the multi-objective fuzzy dominance-based bacterial foraging algorithm is proposed to solve the economic emission dispatch problem; however, it was not compared with other multi-objective algorithms, and it was difficult to reflect the superiority of the improved algorithm [9]. A multi-objective optimization algorithm based on time-varying accelerated particle swarm optimization has been introduced to the cogeneration problem with a high convergence rate and solving the uncertainty of energy demand and supply for intermittent renewable energy sources, but the method has only been applied to a 7-unit test system and there is a lack of research on other scale test systems [10]. Additionally, an improved large-scale symbiotic organism search algorithm has been proposed to address ED problems with valve point effects, demonstrating an increased ability to identify stable and high-quality solutions in a reasonable amount of time [11]. By combining stochastic exploratory search and learning strategies, an improved gray wolf optimization (GWO) algorithm is designed to solve the ED problem in different dimensions and effectively reduces the generation cost compared with other algorithms [12,13]. In order to consider the impact of wind power generation, a novel chaotic quantum genetic algorithm was developed to solve the ED problem of wind power generation with good results [14]. An improved competitive group optimization algorithm has been proposed to handle both static and dynamic ED problems and minimize the total fuel cost by determining the intra-regional generation and inter-regional power exchange, and has advantages in terms of solution accuracy and reliability compared with other algorithms [15]. Furthermore, an improved fireworks algorithm has been applied to multi-regional ED problems, featuring a new constraint-handling scheme that corrects potential solutions within the feasible search space [16,17]. To address multi-regional ED problems, a chaotic artificial bee colony algorithm has been introduced. This algorithm employs the modified sub-gradient method (MSG) to convert the constrained problem into an unconstrained one, and a resulting improved harmony search algorithm (HSA) has been designed for ED problems [18–20].

Economic dispatch includes static economic dispatch and dynamic economic dispatch. Static economic dispatch assumes constant load, which is difficult to adapt to the dynamic changes of actual power load. Dynamic economic dispatch generally divides a day into 24 h and optimizes daily economic dispatch according to daily load forecast, which is more in line with the actual power system operation. The mainstream of current research is dynamic economic dispatch. The integration of plug-in electric vehicles (PEVs) into DED problems has gained attention with the rise of new energy vehicles [21–26]. Connecting PEVs to the grid (V2G) can help mitigate the fluctuations of the grid and achieve peak clipping and valley filling. For large-scale PEVs, a distributed access and central management approach is generally adopted. The PEVs dispatching center can exchange energy and information directly with the power grid, and then the PEVs dispatching center directs the energy exchange of each vehicle according to the actual situation of the serviceable PEVs, so that orderly charging and discharging coordinated management can be realized. In addition, the conversion speed of PEVs charging and discharging is very fast, so it can

be seen as a distributed energy storage device that can be charged and discharged, and the real-time power flow between PEVs and the grid can be realized through bi-directional power conversion technology, which is also known as electric vehicle to grid (V2G). By accurately transmitting information in both directions between the dispatch center and the PEVs, it is possible to achieve bi-directional, real-time, and controllable energy conversion. Of course, this involves the integration of various technologies such as power electronics, communication, power scheduling, and load forecasting. Therefore, it is possible to control the PEVs for orderly discharging when in the peak period of electricity consumption and control the PEVs for orderly charging when in the low peak period of electricity consumption, so as to reduce the burden of the power grid to a certain extent, improve the safety of the power grid, and achieve the purpose of coordinated charging and discharging.

Compared with the ordinary large-scale energy storage, the use of PEVs as a buffer for the grid is due to the fact that electricity can flow in both directions between the grid and PEVs. Moreover, as the number of PEVs increases, it is necessary to rationalize their management, otherwise the grid will be overloaded when the number of PEVs reaches a certain scale. Despite its potential benefits, the DED problem with PEVs has received limited attention in the literature. The moth flame optimization (MFO) algorithm was introduced by Seyedali Mirjalili in 2015 [27]. It is a swarm intelligence optimization algorithm inspired by the flight patterns of moths, and has been widely used for its simple structure, few parameters, and high efficiency. In recent years, various meta-heuristic algorithms, such as the dragonfly algorithm (DA) [28], multi-verse optimizer (MVO) [29], sine cosine algorithm (SCA) [30], ant line optimizer (ALO) [31], grasshopper optimization algorithm (GOA) [32], salp swarm algorithm (SSA) [33], whale optimization algorithm (WOA) [34], marine predator algorithm (MPA) [35], grey wolf optimizer (GWO) [36], binary bat algorithm (BBA) [37], teaching-learning based optimization (TLBO) [38], have been proposed and applied to the DED problem. However, the application of the MFO algorithm to solve the DED problem with PEVs remains limited.

The remainder of this paper is organized as follows. The DED problem integrating with PEVs formulation is presented in Section 2. The proposed CMFO algorithm is described in detail in Section 3. The performance of CMFO is validated on benchmark problems [39,40], as well as DED cases with PEVs in Section 4. The experimental results are analyzed in Section 5. Finally, the summary of the paper and future work directions are given in Section 6.

## 2. Problem Formulation

The target of DED integrating with PEVs is to determine the optimal generation levels of all online units and PEVs during a specified period of time [41,42], so as to minimize both the fuel cost of thermal power plants and grid fluctuation simultaneously for a given load demand while satisfying various constraints [43,44].

### 2.1. Objective Function

Considering that the steam valve of the steam turbine is suddenly open and it would have a certain impact on the energy consumption of the units, which is the so-called valve point loading effect, the objective function of total fuel cost is given by:

$$\min f_1(P) = \sum_{t=1}^{T} \sum_{i=1}^{N} a_i + b_i P_{t,i} + c_i P_{t,i}^2 + |e_i \sin[f_i(P_i^{\min} - P_{t,i})]| \tag{1}$$

where $f_1(P)$ is the total fuel cost of the thermal power units, $T$ is the number of periods in a scheduling cycle, $N$ is the total number of generator units, $a_i$, $b_i$, $c_i$, $e_i$ and $f_i$ are cost coefficients of $i$th the unit, $P_{t,i}$ is the output power of the $i$th unit at time $t$, and $P_i^{\min}$ is the minimum output power of the $i$th unit.

Furthermore, the large fluctuation in power generation could cause damaging impacts on the power grid. So, V2G is introduced to fill the valley and clip peak, which is formulated as:

$$\min f_2(P, P_{PEV}) = \sum_{t=1}^{T} [\sum_{i=1}^{N} (P_{t+1,i} + P_{PEV,t+1} - P_{L,t+1}) - \sum_{i=1}^{N} (P_{t,i} + P_{PEV,t} - P_{L,t})]^2 \quad (2)$$

where $f_2(P, P_{PEV})$ is to minimize grid fluctuations, $P_{PEV,t}$ is the exchange power between the electric vehicle and the grid at time t, and $P_{L,t}$ is transmission loss at time t.

Here, the objective functions $f_1$ and $f_2$ are combined into $f$ using a weighting factor $\omega$, which can be expressed by:

$$f = f_1 + \omega f_2 \quad (3)$$

*2.2. Constraints*

2.2.1. Power Capacity Constraint

The power outputs of generation units are determined by the physical characteristics of the unit, which should be within the capacity of each specific generation unit:

$$P_i^{\min} \le P_{t,i} \le P_i^{\max} \quad (4)$$

where $P_i^{\min}$ and $P_i^{\max}$ are the lower and upper power limits of the *i*th generator, respectively.

2.2.2. Ramp-Rate Limits Constraint

Due to the inertia of thermal power units, the power outputs cannot dramatically change between two adjacent intervals and are subject to the ramp rate limits, which is useful for extending the service life of the units and given as:

$$\begin{cases} P_{t,i} - P_{t-1,i} \le UR_i \\ P_{t-1,i} - P_{t,i} \le DR_i \end{cases} \quad (5)$$

where $UR_i$ is the ramp-up rate limit and ramp-down rate limit of the *i*th unit, respectively.

2.2.3. Electric Vehicle Constraint

Electric vehicle battery state of charge constraint is described as:

$$SOC_{\min} \le SOC_t \le SOC_{\max} \quad (6)$$

where $SOC_{\min}$ is the lower limit of battery capacity, $SOC_{\max}$ is the upper limit of battery capacity, and $SOC_t$ is the capacity of the battery at time t.

The remaining battery power constraint [21,22] is given by:

$$SOC_t = SOC_{t-1} + \eta_{PEV} P_{PEVI,t} \Delta t - \frac{1}{\eta_{PEVO}} P_{PEVO,t} - SOC_{Used,t} \quad (7)$$

where $SOC_t$ is the remaining capacity of the battery at time $t$, $\eta_{PEV}$ and $\eta_{PEVO}$ are the charging efficiency and discharge efficiency of the electric vehicle [23], and $SOC_{Used,t}$ is the amount of electricity that the electric vehicle has used at time t.

Electric vehicle charging and discharging power constraints [24] follow the expression:

$$\begin{cases} P_{PEV,disc}^{\max} \le P_{PEV,t} \le P_{PEV,char}^{\max} \\ \sum_{t=1}^{T} P_{PEV,t} \le P_{PEV,total} \end{cases} \quad (8)$$

where $P_{PEV,disc}^{\max}$ is the maximum discharge power of the electric vehicle, $P_{PEV,char}^{\max}$ is the maximum charging power of the electric vehicle, and $P_{PEV,total}$ is the total power of the electric vehicle.

2.2.4. Power Balance Constraint

In order to maintain power balance [25,26], the output power of all units must be equal to the sum of various demands and the transmission loss at each time interval $t$, which is defined as:

$$\sum_{t=1}^{T} P_{t,i} = P_{D,t} + P_{L,t} + P_{PEV,t} \tag{9}$$

where $P_{D,t}$ is the load demand at time $t$, $P_{L,t}$ is the transmission loss at time $t$, and its mathematic model is formulated as:

$$P_{L,t} = \sum_{i=1}^{N}\sum_{j=1}^{N} P_{t,i} B_{ij} P_{t,j} + \sum_{i=1}^{N} B_{0i} P_{t,i} + B_{00} \tag{10}$$

where $B_{ij}$, $B_{0i}$, and $B_{00}$ are the network loss coefficients.

2.3. *Determination of the Generation Level of the Slack Generator*

As seen from Equation (10), the transmission loss of the power grid is the function of the output power of the units. The slack generator is usually used to decompose $P_{L,t}$, and the decision variable $P_{t,N}$ is separated. Equation (10) could be expressed as:

$$P_{L,t} = B_{NN} P_{t,N}^2 + (2\sum_{i=1}^{N-1} B_{N,i} P_{t,i} + B_{N0}) P_{t,N} + (\sum_{i=1}^{N-1}\sum_{j=1}^{N-1} P_{t,i} B_{ij} P_{t,j} + \sum_{i=1}^{N-1} B_{0i} P_{t,i} B_{00}) \tag{11}$$

Similarly, Equation (9) could be transformed into the following equation:

$$P_{t,N} = P_{D,t} + P_{L,t} + P_{PEV,t} - \sum_{i=1}^{N-1} P_{t,i} \tag{12}$$

Combining Equations (11) and (12), the output power of the slack generator $P_{t,N}$ can be obtained as:

$$B_{NN} P_{t,N}^2 + (2\sum_{i=1}^{N-1} B_{N,i} P_{t,i} + B_{N0} - 1) P_{t,N} + (P_{D,t} + \sum_{i=1}^{N-1}\sum_{j=1}^{N-1} P_{t,1} B_{ij} P_{t,j} + \sum_{i=1}^{N-1} B_{i0} P_{t,i} - \sum_{i=1}^{N-1} P_{t,i} + B_{00}) = 0 \tag{13}$$

## 3. The Modified Moth Flame Algorithm

### 3.1. Brief Overview of MFO

The moth flame optimization (MFO) algorithm is a novel swarm intelligence optimization algorithm proposed by scholar Seyedali Mirjalili in 2015 [27]. It is inspired by the spiral flight of moths at night, as they follow the moon and adjust their flight direction accordingly. However, the artificial flame is very close compared to the moon. Maintaining a fixed angle with the artificial light would eventually generate a spiral flight path approaching the flame for moths. The MFO has strong parallel optimization ability and can explore a wide range of solution spaces. For non-convex problems such as DED with a large number of local optimal points, the MFO is more suitable. Figure 1 depicts the flowchart of the MFO algorithm.

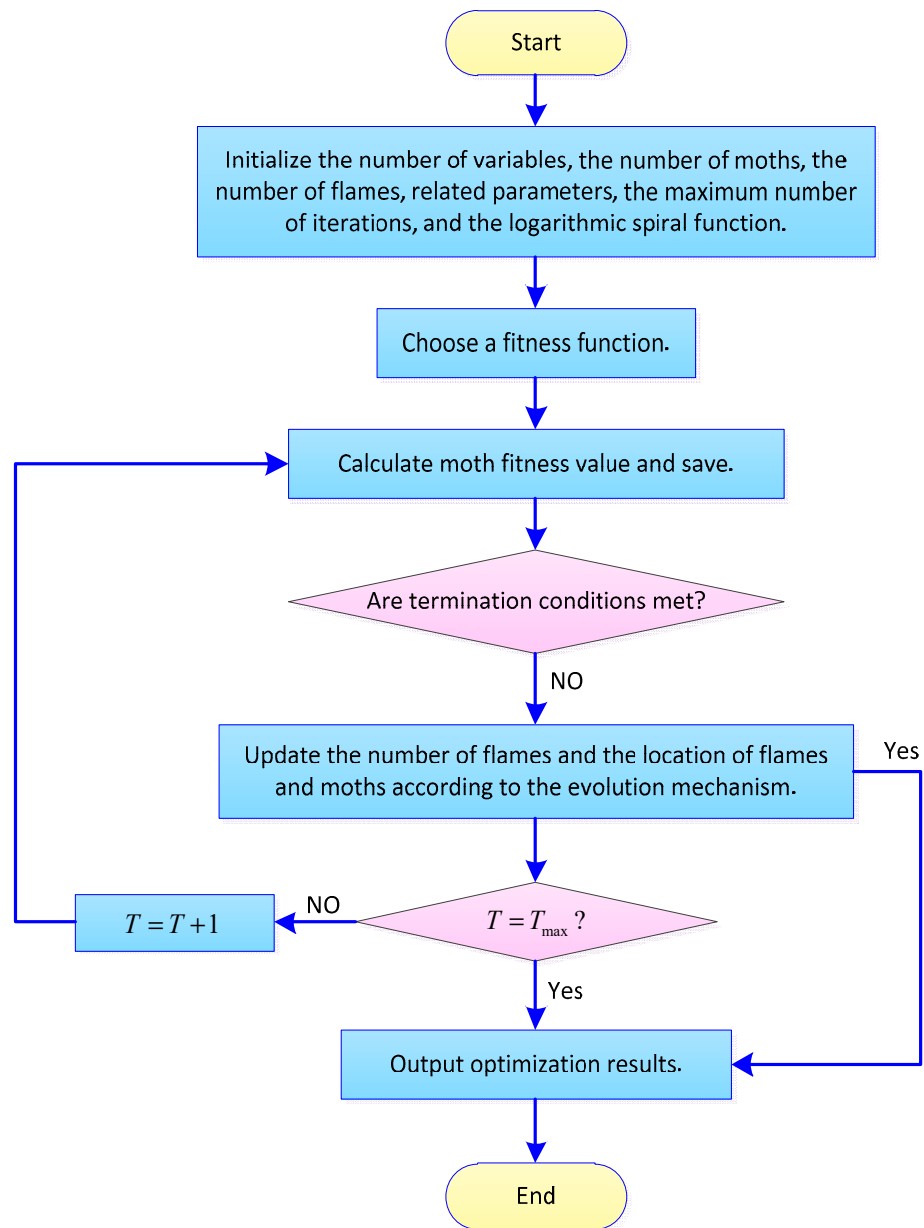

**Figure 1.** Flowchart of MFO.

### 3.1.1. Initialize Parameters

The MFO is essentially a swarm intelligence optimization algorithm. For the ED problem, the candidate solutions are denoted by $m$, that is, the moths are denoted by $m$. Moths fly in a one-dimensional or multi-dimensional manner in the feasible domain, and their flight paths are the range of the solution. The moths' population $M$ is described as:

$$M = \begin{bmatrix} m_{1,1} & m_{1,2} & \cdots & m_{1,d} \\ m_{2,1} & m_{2,2} & \cdots & m_{2,d} \\ \vdots & \vdots & \ddots & \vdots \\ m_{n,1} & m_{n,2} & \cdots & m_{n,d} \end{bmatrix} \tag{14}$$

where $n$ is the number of moths, and $d$ is the size of the dimension.

In the MFO, each moth has a corresponding flame, and the moth flies along its corresponding flame to update its position. The moth corresponds with the flame represented as *F* by the dimension. The position of the flame is expressed as:

$$F = \begin{bmatrix} F_{1,1} & F_{1,2} & \cdots & F_{1,d} \\ F_{2,1} & F_{2,2} & \cdots & F_{2,d} \\ \vdots & \vdots & \ddots & \vdots \\ F_{n,1} & F_{n,2} & \cdots & F_{n,d} \end{bmatrix} \tag{15}$$

The upper bound *ub* and lower bound *lb* in the search space can be given as follows:

$$ub = [ub_1, ub_2, ub_3, \cdots\cdots, ub_{n-1}, ub_n] \tag{16}$$

$$lb = [lb_1, lb_2, lb_3, \cdots\cdots, lb_{n-1}, lb_n] \tag{17}$$

### 3.1.2. The Moth's Location Updating

The moth flies following the logarithmic spiral function *S*, which is constructed as:

$$S(M_i, F_i) = D_i e^{bt} \cos(2\pi t) + F_j \tag{18}$$

where *b* is the parameter of the logarithmic spiral shape, and $t \in [-1, 1]$ is the distance parameter. The value *t* is proportionate to the distance of the moth relative to the flame. If $t = 1$, the moth is far away from the flame, and if $t = -1$, the distance of the moth relative to the flame is very short, so the flying domain of the moth is the entire global space, including the flame itself and the space occupied by the flame. The variable *K* can be introduced into the value of the path coefficient *t*, where $K \in [-1, -2]$. Therefore, the value range of *t* is $t \in [K, 1]$. As the number of iterations increases, the value of *K* changes linearly from big to small, which improves the efficiency of moths approaching the flames.

$D_i$ is the distance from the *i*th moth to the *j*th flame and formulated as:

$$D_i = \left| F_j - M_i \right| \tag{19}$$

where $F_j$ is the *j*th flame. The number of flames *flame_no* needs to be adaptively updated to reduce calculation time and improve operating efficiency, which is updated by:

$$flame\_no = round(N - k \times \frac{N - 1}{T}) \tag{20}$$

where *k* is the current iteration number, *N* is the maximum number of flames, and *T* is the maximum number of iterations.

The position update mechanism of the MFO is critical to unlocking its full local search capabilities. So, the optimal position generated by the moth in the previous iteration should be updated in the current iteration, meaning that if the fitness value is better than the flame, the position of the flame is updated. In each iteration, the position of the flame is saved in the matrix *F*, and then the moth updates the position according to the matrix *F*; the update mechanism $M_i$ is expressed as:

$$M_i = S(M_i, F_j) \tag{21}$$

### 3.2. Chaos Moth Flame Algorithm

The chaos moth flame (CMFO) algorithm is an enhanced version of the MFO algorithm [27]. It integrates chaotic mapping with the evolutionary mechanism of the MFO to produce improved results. The CMFO algorithm uses a chaotic sequence, generated through chaotic mapping, to initialize the solution space and enhance the search process. As a result, the CMFO algorithm is able to obtain better initial solutions compared with the MFO algorithm. The pseudo-code for the CMFO algorithm is presented in Algorithm 1.

| **Algorithm 1.** Chaos moth flame optimization | |
|---|---|
| **Input:** | Population size $NP = 30$. |
| **Output:** | The final solution $X_{best}$ and its fitness $f_{best}$. |
| 1: | Set the iteration $FEs = 1$ and maximum iteration $Max\_FEs$; |
| 2: | **for** $i = 1$ *to dim* |
| 3: |   **for** $j = 1$ *to NP* |
| 4: |   Introduce chaotic mapping using Equation (23); |
| 5: |   Initialize the upper boundary and the lower boundary $(ub, lb)$; |
| 6: |   Initialize the position of the *i*th particle $X_{ij}$ and use mapping; |
| 7: |   **end for** |
| 8: | **end for** |
| 9: | **While** $FES <= Max\_FES$ *do* |
| 10: |   Adaptively update the number of $flame\_no$ using Equation (20); |
| 11: |   **for** $i = 1$ *to dim* |
| 12: |   Check if moths go out of the search space through $(ub, lb)$ and bring it back; |
| 13: |   Calculate the fitness of moths $f(X_{ij})$; |
| 14: |   **end for** |
| 15: |    **if** $FES == 1$ |
| 16: |   Sort the first population of moths $f(X_{best})$; |
| 17: |   Update the flames $best\_flame\_ f(X_{ij})$; |
| 18: |   **else** |
| 19: |   Re-combinate the moth and flame; Calculate the $double\_fitness(moth\_f(X_{ij}), flame\_f(X_{ij}))$; |
| 20: |   Sort the re-combinate population $double\_fitness$; |
| 21: |   Update the flames $flame\_f(X_{best})$ using Equation (21); |
| 22: |   **end if** |
| 23: |   Calculate parameter a using the relevant formula; |
| 24: |   **for** $i = 1$ *to dim* |
| 25: |   Calculate parameter b using chaotic mapping through Equation (23); |
| 26: |    **for** $j = 1$ *to NP* |
| 27: |   **if** $i <= Flame\_no$ |
| 28: |    Calculate the distance between the moth and the flame $D_{ij}$ using Equation (19); |
| 29: |    Calculate the path coefficient $t$ using the relevant formula; |
| 30: |    Update the position $X_{ij}$ using Equation (18), Equation (23); |
| 31: |   **end if** |
| 32: |   **if** $i > Flame\_no$ |
| 33: |    Calculate the distance between the moth and the flame $D_{ij}$ using Equation (19); |
| 34: |    Calculate the path coefficient $t$ using the relevant formula; |
| 35: |     Update the position $X_{ij}$ using Equation (21), Equation (23); |
| 36: |   **end if** |
| 37: |   **end for** |
| 38: |   **end for** |
| 39: |   Update the global best position $X_{best}$ and its fitness $f_{best}$; $FES = FES + 1$; |
| 40: | **end while** |

Chaos Mapping

Chaos is a complex and ubiquitous phenomenon in non-linear deterministic systems, where it showcases the interplay between order and disorder, determinism, and randomness [7]. Chaos is limited to a specific region, is unique and never repeats, and its orbit is complex. Although it appears random, chaos is in fact a deterministic system [14], where the state of the system at any moment is influenced by its previous state. This makes chaos unpredictable and seemingly random, yet it also exhibits good autocorrelation and low-frequency broadband, making it distinct from periodic and quasi-periodic motion. Chaos is characterized by its extreme sensitivity to initial conditions, which is one of its most prominent features. The chaotic map is described by different probability distributions that can be quantified using a probability density function. In order to measure the chaotic

degree [17] of the dynamic system, *LE* is introduced, which is used to evaluate the chaotic degree of the system as follows:

$$LE = \lim_{n \to \infty} \frac{1}{n} \sum_{k=1}^{n} \ln |f'(x_k)| \tag{22}$$

For all one-dimensional chaotic maps, $LE > 0$, the three chaotic maps introduced are sine map, iterative map, chebyshev map, and correlation chaos map in Table 1.

**Table 1.** Chaotic maps.

| NO. | Chaotic Map Equations | LE |
|---|---|---|
| 1. | Sine map: $\begin{cases} X_{K+1} = (a/4) \cdot \sin(\pi X_k) \\ 0 < a \leq 4, a = 4, X_k \in (0,1) \end{cases}$ | 0.6885 |
| 2. | Iterative map: $\begin{cases} X_{K+1} = \sin(a/X_k) \\ a \in (0,\infty), a = 2, X_k \in [-1,1]/\{0\} \end{cases}$ | 1.6556 |
| 3. | Chebyshev: $\begin{cases} X_{K+1} = \cos(a \cdot \cos^{-1} X_k) \\ a > 0, a = 3, X_k \in [-1,1] \end{cases}$ | 1.0986 |

Since the parameters of the logarithmic helix are fixed, the shape of the logarithmic helix is also fixed for MFO [27], which makes the local search ability worse. So, MFO deals with problems with high dimensionality and non-convex, which tend to get stuck in local optima. Here, the chaotic map is introduced to the evolution mechanism of the logarithmic spiral, which makes the parameters of the logarithmic spiral change in a chaotic shape in the iteration process of the CMFO. So, it makes CMFO search more comprehensively in the solution space and has strong global exploration ability. In addition, the chaotic map sequence is applied to the initialization process of CMFO, so that the CMFO has a better initial solution, which can reduce the running time of the algorithm and accelerate the convergence speed. The search principles of MFO and CMFO on the solution space are shown in Figure 2.

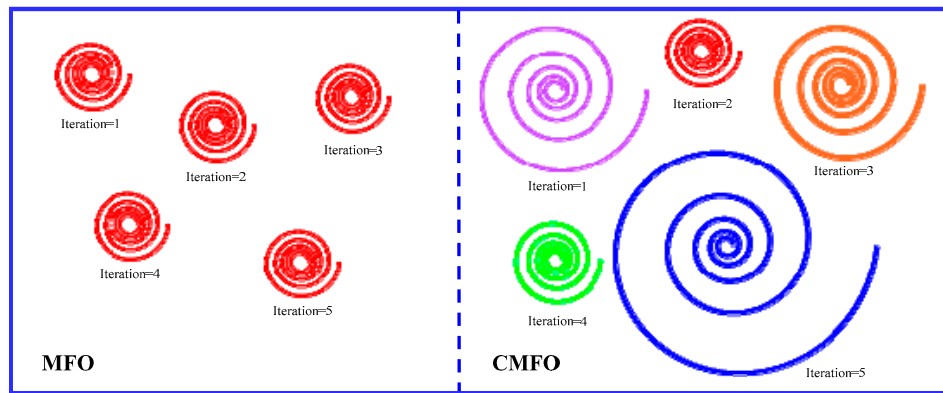

**Figure 2.** Comparison of MFO and CMFO search principle.

In Figure 2, for MFO [27], the moth needs to construct a logarithmic spiral by Equation (18) during the search process. Among the parameters of the logarithmic spiral function, the parameter *b* determines the shape of the logarithmic spiral and is a static parameter. As can be seen from the MFO of Figure 2, the static parameter *b* is weakly adaptive, causing the MFO to poorly explore the solution space.

The chaotic map is applied to the logarithmic spiral evolution mechanism, which is formulated by Equation (23), so that the parameter *b* can change in a chaotic state with the iteration. The parameter *b* is improved to a dynamic chaotic parameter. Taking the two-dimensional solution space as an example, it can be seen from the CMFO of Figure 2 that the application of the dynamic chaos parameter makes the shape of the helix change

continuously with the iterations, which is highly adaptive and more exploratory to the solution space. Therefore, CMFO can well-compensate for the deficiencies of MFO.

In addition, to make the CMFO better converge to the optimal solution, the chaotic mapping sequence is used to generate the initial solution, thereby saving the running time of the CPU. Meanwhile, sine chaotic mapping is used for updating the parameter *b*. The mathematical model of the logarithmic spiral search of CMFO is described as:

$$\begin{cases} S(M_i, F_i) = D_i e^{b_i t} \cos(2\pi t) + F_j \\ b_{i+1} = \sin(\pi b_i) \end{cases} \tag{23}$$

## 4. Performance Verification of CMFO

In order to verify the rationality and validity of the CMFO, it is compared with nine other meta-heuristic algorithms, of which the abbreviations and parameters are shown in Tables 2 and 3, respectively. The above experiments are performed by conducting on seven well-known benchmark functions, which are **F1 (Sphere)**, **F2 (Schwelfel's 2.22)**, **F3 (Schwelfel's 1.2)**, **F4 (Step)**, **F5 (Griewank)**, **F6 (Ackley)**, **F7 (Penalized-1)**, as shown in Table 4, which include unimodal and multimodal functions. Because of only one optimum, the solution of unimodal function can be determined easily. However, for the multimodal function, the number of local minimum increases with the dimension of problems, thus it is difficult to obtain optimal solutions. Therefore, the selected benchmark functions can effectively evaluate the performance of the algorithm in terms of escaping from the local optimum and convergence speed. Furthermore, three dimensions, namely 10, 20, and 50, are selected to test the generalization ability of the algorithm, and each trial runs 30 times independently. For the test case of 10 dimensions and 20 dimensions, the number of iterations is 1000. For the test case of 50 dimensions, the number of iterations is 10,000. All experiments in this study are conducted in a PC with **MATLAB** ®**2020a**, **Intel(R) Core (TM) i5-7200U CPU @ 2.50 GHz(4CPUs)**, and **RAM 8.0 GHz**. The experiment results are shown in Tables 5–7, which are in terms of the standard deviation **"Sta"**, optimal value **"Best"** of the best-so-far solution, and average CPU running time **"Time(s)"**.

**Table 2.** List of algorithm abbreviations.

| Table of Abbreviations | | Abbreviation |
|---|---|---|
| 1. | Moth-Flame Optimization Algorithm | MFO [27] |
| 2. | Dragonfly Algorithm | DA [28] |
| 3. | Multi-Verse Optimizer | MVO [29] |
| 4. | Sine Cosine Algorithm | SCA [30] |
| 5. | Ant Lion Optimizer | ALO [31] |
| 6. | Grasshopper Optimisation Algorithm | GOA [32] |
| 7. | Salp Swarm Algorithm | SSA [33] |
| 8. | Whale Optimization Algorithm | WOA [34] |
| 9. | Ocean Predator Algorithm | MPA [35] |
| 10. | Grey Wolf Optimizer | GWO [36] |
| 11. | Chaos Moth-Flame Optimization Algorithm | CMFO |

**Table 3.** The parameters of the algorithms.

| Algorithms | Parameter Settings | Iteration |
|---|---|---|
| MFO [27] | NP = 30 | 1000, 10,000 |
| DA [28] | NP = 30, Beta = 1.5 | 1000, 10,000 |
| MVO [29] | NP = 30, WEP_Max = 1, WEP_min = 0.2 | 1000, 10,000 |
| SCA [30] | NP = 30, $a = 2$ | 1000, 10,000 |
| ALO [31] | NP = 30 | 1000, 10,000 |
| GOA [32] | NP = 30, Cmax = 1, Cmin = 0.00004 | 1000, 10,000 |
| SSA [33] | NP = 30 | 1000, 10,000 |
| WOA [34] | NP = 30, $b = 1$ | 1000, 10,000 |
| MPA [35] | NP = 30, Fads = 0.2, P = 0.5, Beta = 1.5 | 1000, 10,000 |
| GWO [36] | NP = 30 | 1000, 10,000 |
| CMFO | NP = 30 | 1000, 10,000 |

**Table 4.** Benchmark functions.

| Function | Dim | Range | $F_{\min}$ |
|---|---|---|---|
| $F_1(x) = \sum\limits_{i=1}^{n} x_i^2$ | 10/20/50 | $[-100, 100]$ | 0 |
| $F_2(x) = \sum\limits_{i=1}^{n} \lvert x_i \rvert + \prod\limits_{i=1}^{n} \lvert x_i \rvert$ | 10/20/50 | $[-100, 100]$ | 0 |
| $F_3(x) = \sum\limits_{i=1}^{n} \left( \sum\limits_{j=1}^{i} x_j \right)^2$ | 10/20/50 | $[-100, 100]$ | 0 |
| $F_4(x) = \sum\limits_{i=1}^{n} \left( [x_i + 0.5] \right)^2$ | 10/20/50 | $[-100, 100]$ | 0 |
| $F_5(x) = \frac{1}{4000} \sum\limits_{i=1}^{n} x_i{}^2 - \prod\limits_{i=1}^{n} \cos\left( \frac{x_i}{\sqrt{i}} \right) + 1$ | 10/20/50 | $[-600, 600]$ | 0 |
| $\begin{cases} F_6(x) = \frac{\pi}{n} \left\{ 10 \sin(\pi y_1) + \sum\limits_{i=1}^{n-1} (y_i - 1)^2 [1 + 10 \sin^2(\pi y_{i+1})] + (y_n - 1)^2 \right\} + \sum\limits_{i=1}^{n} u(x_i, 10, 100, 4) \\ \qquad\qquad\qquad\qquad y_i = 1 + \frac{x_i + 1}{4} \end{cases}$ | 10/20/50 | $[-50, 50]$ | 0 |
| $F_7(x) = 0.1 \{ \sin^2(3\pi x_i) + \sum\limits_{i=1}^{n} (x_i - 1)^2 [1 + \sin^2(3\pi x_i + 1)] + (x_n - 1)^2 [1 + \sin^2(2\pi x_n)] \} + \sum\limits_{i=1}^{n} u(x_i, 5, 100, 4)$ | 10/20/50 | $[-50, 50]$ | 0 |

In Tables 5, A1 and A2, in Appendix A, among the three metrics "Sta", "Best", and "Time", N means that the algorithm is worse than the CMFO in more than two metrics, and Y means that the algorithm is better than the CMFO in more than two metrics. Furthermore, '+' means that the performance of the algorithm is superior. Tables 5, A1 and A2, in Appendix A, show that the proposed CMFO is better than the other algorithms except for WOA [34], MPA [35], and GWO [36], according to the statistical results. In particular, the small optimal value and standard deviation of different kinds of functions indicate that the CMFO has high solution precision and stability. Furthermore, compared with other methods, the CMFO significantly reduces the time of computing. For example, in 10 dimensions, the CMFO converged 305%, 916%, 1511%, 135%, 2%, 163%, 24%, 147%, 77%, and 9% faster compared with the other 10 algorithms in the unimodal test function F1, and also improved the stability of the optimal values. In the multimodal test function F6, the CMFO improves the convergence speed compared with the other 10 algorithms by 1544%, 314%, 589%, 52%, −3%, 104%, −1%, 110%, 18%, and 7%, respectively, and improves the stability and accuracy of the optimal values. In the multimodal test function F7, the CMFO converges 102%, 339%, 684%, 62%, 9%, 121%, 8%, 121%, 38%, and 4% faster than the other 10 algorithms, respectively, and also improves the accuracy of the optimal values. Moreover, the advantages of CMFO in optimizing unimodal and multimodal test functions remain when improved to 20 and 50 dimensions, indicating that the improved algorithm is equally suitable for solving high-dimensional, high-complexity optimization problems, making it a good choice to use CMFO for problems with higher dimensionality and complexity like DED.

In order to provide an intuitive comparison, the convergence curves of **Step**, **Ackley**, and **Penalized-1** representatives are illustrated in Figures 3 and A1 in Appendix A. It can be seen from the convergence curves that the CMFO has stronger exploration ability, faster convergence, more generality, and higher accuracy compared with other algorithms. Such improvements are related to chaotic mapping and are introduced into CMFO. Meanwhile, the experiment results fully demonstrate that the proposed CMFO is promising and competitive.

**Table 5.** Comparison of CMFO with other algorithms (10 dimensions).

| NO. | Statistics | ALO | DA | GOA | MVO | SSA | WOA | SCA | MPA | GWO | MFO | CMFO |
|---|---|---|---|---|---|---|---|---|---|---|---|---|
| **F1** | **Sta** | $1.40 \times 10^{-9}$ | $2.34 \times 10^{1}$ | $1.50 \times 10^{-7}$ | $1.80 \times 10^{-3}$ | $3.30 \times 10^{-10}$ | $4.24 \times 10^{-155}$ | $4.74 \times 10^{-25}$ | $2.93 \times 10^{-63}$ | $9.89 \times 10^{-133}$ | $9.46 \times 10^{-22}$ | $2.06 \times 10^{-34}$ |
| | **Best** | $8.42 \times 10^{-10}$ | $2.50 \times 10^{-3}$ | $6.37 \times 10^{-8}$ | $1.20 \times 10^{-3}$ | $4.13 \times 10^{-10}$ | $1.70 \times 10^{-175}$ | $3.69 \times 10^{-34}$ | $2.89 \times 10^{-65}$ | $4.91 \times 10^{-123}$ | $1.30 \times 10^{-20}$ | $1.55 \times 10^{-36}$ |
| | **Time(s)** | $1.45 \times 10^{1}$ | $3.64 \times 10^{1}$ | $5.77 \times 10^{1}$ | $8.44 \times 10^{-1}$ | $3.65 \times 10^{-1}$ | $9.44 \times 10^{-1}$ | $4.45 \times 10^{-1}$ | $8.86 \times 10^{-1}$ | $6.34 \times 10^{-1}$ | $3.91 \times 10^{-1}$ | $3.58 \times 10^{-1}$ |
| | **Winner** | N | N | N | N | N | Y | N | Y | Y | N | + |
| **F2** | **Sta** | $1.34 \times 10^{0}$ | $1.12 \times 10^{0}$ | $8.08 \times 10^{1}$ | $1.16 \times 10^{-2}$ | $9.19 \times 10^{-6}$ | $9.27 \times 10^{-106}$ | $1.20 \times 10^{-18}$ | $5.90 \times 10^{-35}$ | $7.86 \times 10^{-67}$ | $1.01 \times 10^{-13}$ | $1.50 \times 10^{-20}$ |
| | **Best** | $1.44 \times 10^{-5}$ | $6.91 \times 10^{-1}$ | $2.01 \times 10^{2}$ | $8.10 \times 10^{-3}$ | $4.58 \times 10^{-6}$ | $3.15 \times 10^{-116}$ | $7.80 \times 10^{-22}$ | $1.13 \times 10^{-36}$ | $7.54 \times 10^{-69}$ | $2.56 \times 10^{-15}$ | $2.84 \times 10^{-22}$ |
| | **Time(s)** | $1.42 \times 10^{1}$ | $3.85 \times 10^{1}$ | $5.99 \times 10^{1}$ | $9.15 \times 10^{-1}$ | $4.09 \times 10^{-1}$ | $1.26 \times 10^{0}$ | $4.56 \times 10^{-1}$ | $9.50 \times 10^{-1}$ | $6.68 \times 10^{-1}$ | $4.51 \times 10^{-1}$ | $4.38 \times 10^{-1}$ |
| | **Winner** | N | N | N | N | N | Y | N | Y | Y | N | + |
| **F3** | **Sta** | $8.10 \times 10^{-6}$ | $6.02 \times 10^{1}$ | $1.02 \times 10^{-1}$ | $1.42 \times 10^{-2}$ | $1.24 \times 10^{-9}$ | $2.86 \times 10^{1}$ | $5.45 \times 10^{-8}$ | $3.07 \times 10^{-31}$ | $1.08 \times 10^{-53}$ | $1.27 \times 10^{-5}$ | $1.10 \times 10^{-10}$ |
| | **Best** | $2.22 \times 10^{-6}$ | $2.19 \times 10^{0}$ | $1.62 \times 10^{-4}$ | $4.80 \times 10^{-3}$ | $6.00 \times 10^{-10}$ | $9.64 \times 10^{-6}$ | $3.68 \times 10^{-13}$ | $3.56 \times 10^{-36}$ | $2.59 \times 10^{-61}$ | $3.97 \times 10^{-3}$ | $3.60 \times 10^{-13}$ |
| | **Time(s)** | $1.42 \times 10^{1}$ | $3.89 \times 10^{1}$ | $6.48 \times 10^{1}$ | $1.05 \times 10^{0}$ | $5.66 \times 10^{-1}$ | $1.31 \times 10^{0}$ | $5.43 \times 10^{-1}$ | $1.33 \times 10^{0}$ | $8.57 \times 10^{-1}$ | $6.01 \times 10^{-1}$ | $5.83 \times 10^{-1}$ |
| | **Winner** | N | N | N | N | N | N | N | Y | Y | N | + |
| **F4** | **Sta** | $6.63 \times 10^{-10}$ | $1.33 \times 10^{1}$ | $2.66 \times 10^{-7}$ | $1.70 \times 10^{-3}$ | $3.16 \times 10^{-10}$ | $8.42 \times 10^{-5}$ | $9.86 \times 10^{-2}$ | $9.99 \times 10^{-15}$ | $1.97 \times 10^{-7}$ | $2.19 \times 10^{-10}$ | $0.00 \times 10^{0}$ |
| | **Best** | $9.24 \times 10^{-10}$ | $1.58 \times 10^{-2}$ | $3.59 \times 10^{-8}$ | $1.20 \times 10^{-3}$ | $2.11 \times 10^{-10}$ | $1.04 \times 10^{-5}$ | $1.70 \times 10^{-1}$ | $4.39 \times 10^{-16}$ | $4.86 \times 10^{-7}$ | $3.92 \times 10^{-17}$ | $0.00 \times 10^{0}$ |
| | **Time(s)** | $1.28 \times 10^{0}$ | $3.62 \times 10^{1}$ | $5.92 \times 10^{1}$ | $9.10 \times 10^{-1}$ | $4.06 \times 10^{-1}$ | $8.38 \times 10^{-1}$ | $4.35 \times 10^{-1}$ | $9.45 \times 10^{-1}$ | $6.17 \times 10^{-1}$ | $3.97 \times 10^{-1}$ | $3.92 \times 10^{-1}$ |
| | **Winner** | N | N | N | N | N | N | N | N | N | N | + |
| **F5** | **Sta** | $9.07 \times 10^{-2}$ | $3.01 \times 10^{-1}$ | $1.19 \times 10^{-1}$ | $9.36 \times 10^{-2}$ | $9.78 \times 10^{-2}$ | $1.42 \times 10^{-1}$ | $9.45 \times 10^{-2}$ | $9.87 \times 10^{-2}$ | $1.19 \times 10^{-1}$ | 4.9748 | $9.04 \times 10^{-2}$ |
| | **Best** | $3.20 \times 10^{-2}$ | $1.51 \times 10^{-1}$ | $1.40 \times 10^{-1}$ | $8.48 \times 10^{-2}$ | $4.93 \times 10^{-2}$ | $0.00 \times 10^{0}$ | $0.00 \times 10^{0}$ | $0.00 \times 10^{0}$ | $0.00 \times 10^{0}$ | 16.1056 | $0.00 \times 10^{0}$ |
| | **Time(s)** | $1.42 \times 10^{1}$ | $3.67 \times 10^{1}$ | $6.47 \times 10^{1}$ | $1.00 \times 10^{0}$ | $4.84 \times 10^{-1}$ | $1.04 \times 10^{0}$ | $4.95 \times 10^{-1}$ | $1.27 \times 10^{0}$ | $6.90 \times 10^{-1}$ | $4.72 \times 10^{-1}$ | $4.91 \times 10^{-1}$ |
| | **Winner** | N | N | N | N | N | N | N | N | N | N | + |
| **F6** | **Sta** | $6.95 \times 10^{-1}$ | $1.35 \times 10^{0}$ | $8.90 \times 10^{-1}$ | $1.43 \times 10^{-1}$ | $2.67 \times 10^{-1}$ | $4.00 \times 10^{-3}$ | $2.54 \times 10^{-2}$ | $2.77 \times 10^{-15}$ | $7.86 \times 10^{-3}$ | 0.2801 | $9.47 \times 10^{-33}$ |
| | **Best** | $2.58 \times 10^{-11}$ | $5.80 \times 10^{-2}$ | $6.07 \times 10^{-6}$ | $1.56 \times 10^{-5}$ | $5.12 \times 10^{-12}$ | $2.28 \times 10^{-5}$ | $3.32 \times 10^{-2}$ | $4.25 \times 10^{-16}$ | $1.20 \times 10^{-7}$ | $2.37 \times 10^{-7}$ | $4.71 \times 10^{-32}$ |
| | **Time(s)** | $1.46 \times 10^{1}$ | $3.68 \times 10^{1}$ | $6.12 \times 10^{1}$ | $1.35 \times 10^{0}$ | $8.62 \times 10^{-1}$ | $1.82 \times 10^{0}$ | $8.78 \times 10^{-1}$ | $1.87 \times 10^{0}$ | $1.05 \times 10^{0}$ | $9.49 \times 10^{-1}$ | $8.88 \times 10^{-1}$ |
| | **Winner** | N | N | N | N | N | N | N | N | N | N | + |
| **F7** | **Sta** | $4.39 \times 10^{-3}$ | $2.18 \times 10^{-1}$ | $1.35 \times 10^{-2}$ | $5.40 \times 10^{-3}$ | $4.40 \times 10^{-3}$ | $2.61 \times 10^{-2}$ | $6.93 \times 10^{-2}$ | $1.18 \times 10^{-13}$ | $3.98 \times 10^{-2}$ | $1.47 \times 10^{-5}$ | $6.92 \times 10^{-32}$ |
| | **Best** | $1.04 \times 10^{-10}$ | $3.56 \times 10^{-2}$ | $4.19 \times 10^{-6}$ | $1.11 \times 10^{-4}$ | $2.12 \times 10^{-11}$ | $7.70 \times 10^{-5}$ | $1.86 \times 10^{-1}$ | $7.23 \times 10^{-15}$ | $1.00 \times 10^{-6}$ | $3.79 \times 10^{-8}$ | $1.35 \times 10^{-32}$ |
| | **Time(s)** | $1.69 \times 10^{1}$ | $3.67 \times 10^{1}$ | $6.55 \times 10^{1}$ | $1.36 \times 10^{0}$ | $9.12 \times 10^{-1}$ | $1.85 \times 10^{0}$ | $9.01 \times 10^{-1}$ | $1.85 \times 10^{0}$ | $1.15 \times 10^{0}$ | $8.71 \times 10^{-1}$ | $8.35 \times 10^{-1}$ |
| | **Winner** | N | N | N | N | N | N | N | N | N | N | + |

**Table 6.** Scenarios.

| Scenario 1: Only Units | Dimensions | Scenario 2: Units with PEVs | Dimensions |
|---|---|---|---|
| **Case I**: 5 units | $5 \times 24 = 120$ | **Case IV**: 5 units + PEVs | $6 \times 24 = 144$ |
| **Case II**: 10 units | $10 \times 24 = 240$ | **Case V**: 10 units + PEVs | $11 \times 24 = 264$ |
| **Case III**: 15 units | $15 \times 24 = 360$ | **Case VI**: 15 units + PEVs | $16 \times 24 = 384$ |

**Table 7.** Comparison of whether PEVs in DED.

| Parameter | Best Cost ($) | Worst Cost ($) | Average Cost ($) | Fluctuation (MW)^2 | Sta | CPU Time (s) |
|---|---|---|---|---|---|---|
| **Scenario 1** | | | | | | |
| **caseI**: 5 units | $3.98 \times 10^4$ | $4.07 \times 10^4$ | $3.99 \times 10^4$ | $1.06 \times 10^6$ | $6.67 \times 10^2$ | $2.65 \times 10^2$ |
| **caseII**: 10 units | $2.28 \times 10^6$ | $2.31 \times 10^6$ | $2.29 \times 10^6$ | $1.97 \times 10^6$ | $4.85 \times 10^4$ | $9.57 \times 10^2$ |
| **caseIII**: 15 units | $6.71 \times 10^5$ | $6.95 \times 10^5$ | $6.74 \times 10^5$ | $2.09 \times 10^6$ | $9.67 \times 10^3$ | $2.05 \times 10^3$ |
| **Scenario 2** | | | | | | |
| **caseIV**: 5 units | $3.93 \times 10^4$ | $4.19 \times 10^4$ | $4.07 \times 10^4$ | $7.90 \times 10^5$ | $8.87 \times 10^2$ | $2.33 \times 10^2$ |
| **caseV**: 10 units | $2.17 \times 10^6$ | $2.33 \times 10^6$ | $2.25 \times 10^6$ | $1.01 \times 10^6$ | $3.90 \times 10^4$ | $9.30 \times 10^2$ |
| **caseVI**: 15 units | $6.50 \times 10^5$ | $6.83 \times 10^5$ | $6.71 \times 10^5$ | $1.16 \times 10^6$ | $9.00 \times 10^3$ | $1.97 \times 10^3$ |

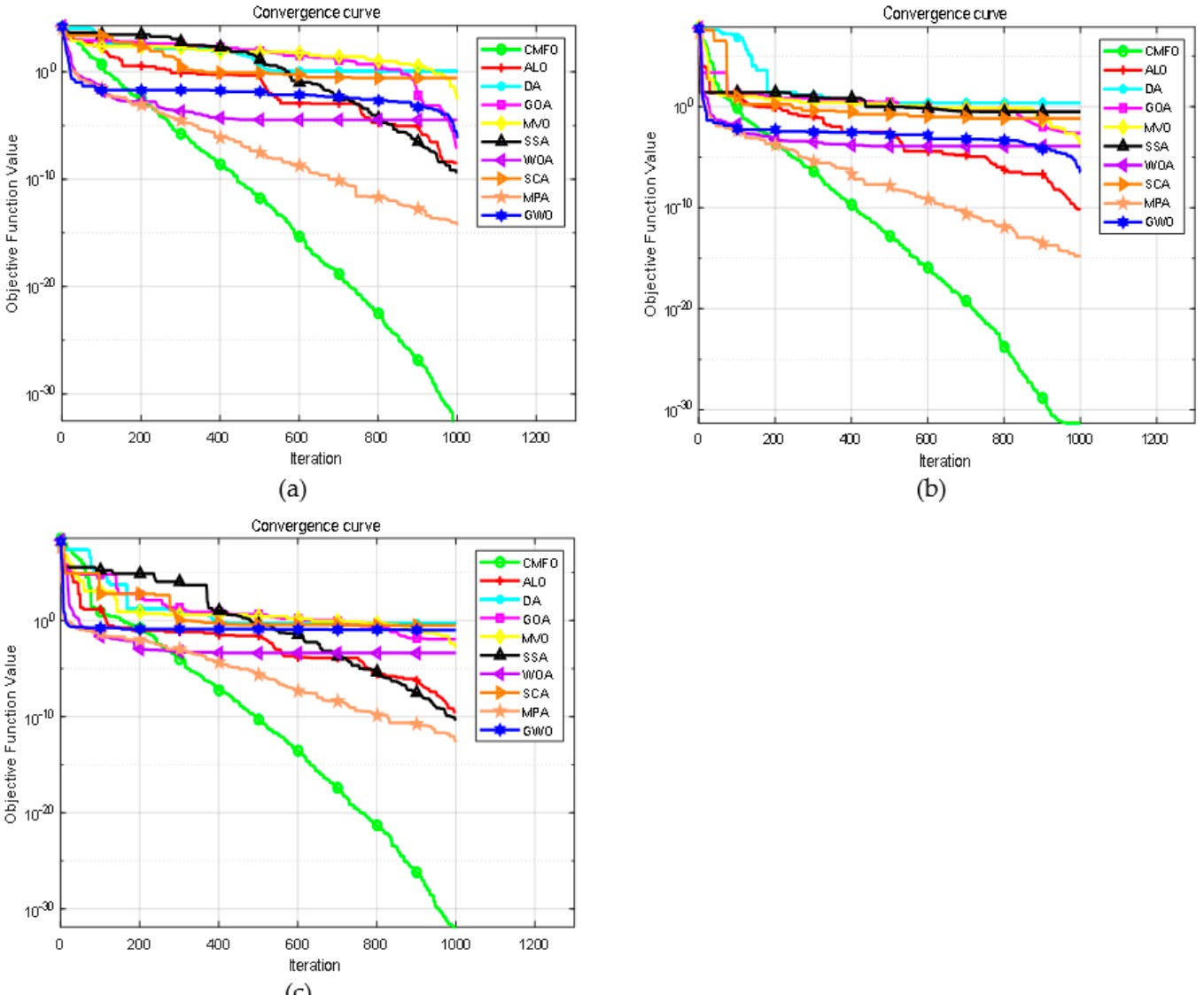

**Figure 3.** The convergence comparison curves of each algorithm. (**a**–**c**): Test functions are Step, Ackley, and Penalized-1, dimension is 10.

## 5. Implementation of CMFO to DED Problems

The conceptual representation of the CMFO approach to solving the dynamic economic dispatch problem with the integration of plug-in electric vehicles is depicted in Figure 4. This representation consists of three integral components: the initial population formation through chaotic mapping, which enhances the diversity of the population; the objective function and constraints of the DED problem, which are transformed from a multi-objective problem to a single-objective problem through weight factors; and the application of the CMFO algorithm to solve the DED problem [45].

Furthermore, the model of plug-in electric vehicles to the grid used in this paper is shown in Figure 5. The system operator determines the day-ahead schedule of thermal power plants according to the power demand as well as coordinates the power delivering/receiving to/from the PEV aggregators [46]. In plug-in electric vehicles to the grid mode, PEV aggregators are designed to possess options for delivering or receiving power to/from the grid. The model is simple and efficient, so it is used to study the DED problem.

### 5.1. Parameter Setting

The population size of the CMFO is **NP = 500**, the number of PEVs is **75,000**, the charging and discharging efficiencies are both **0.85**, the available PEVs are **20%**, and the average battery capacity is **0.015 MW**. The **V2G** error is **0.01**, and the objective function weight value is set to **1**. In addition, the charging and discharging power of PEVs is 100 MW, and for 5 units, the maximum power generation of a single unit is 300 MW; the minimum value of power generation demand for each time period is 410 MW and the maximum value is 720 MW. For 10 units, the maximum generation power of each single unit is 470 MW, and the minimum value of generation demand for each time period is 1036 MW and the maximum value is 1972 MW. For 15 units, the maximum generation power of each single unit is 470 MW, and the minimum value of generation demand for each time period is 1171 MW and the maximum value is 2394 MW. Therefore, from the power generation perspective, the power ratios of a single genset and a PEV are 3:1, 4.7:1, and 4.7:1, respectively, and the impact of a PEV is 13.9~24.4% for 5 units, 5.1~9.7% for 10 units, and 4.2~8.5% for 15 units. Therefore, the reasonable deployment of electric vehicle charging and discharging can reduce the grid fluctuation to a certain extent and achieve the effect of peak and valley reduction.

Regarding the algorithms used for comparison, for MFO [27], **NP = 500** and the crossover probability **PC = 0.7**. For GADMFI [26], the mutation probability **Pm = 0.3** and the population size **NP = 100**. For MPA [35], **NP = 500**, **Fads = 0.2**, **P = 0.5**, and **Beta = 1.5**. The number of iterations for each algorithm remains fixed. In order to verify the reliability of the proposed algorithm for DED problems, two scenarios and six cases are considered, as described in Table 6.

### 5.2. Scenarios 1 and 2

The three cases of Scenario 1 are power systems with 5 units, 10 units, and 15 units, respectively, which are used to test the performance of the CMFO. For Scenario 2, PEVs are connected to the grid to reduce the fluctuation of the power grid, and the charging mode is random charging, which is more in line with the actual situation, regardless of forcing the user to choose to discharge at peak power consumption and charge at low power consumption [47,48]. Three cases, **5 units + PEVs**, **10 units + PEVs**, and **15 units + PEVs**, are used to test the performance of the CMFO. The experiment results of **Scenarios 1 and 2** are shown in Table 7. For intuitive analysis of **Units + PEVs**, the output power in each period is shown in Figure 6, and the power curve in each period is shown in Figure 7. In Figure 6, the "FEVs" curve represents the charge and discharge of PEVs in each period, and the "Original Demand + FEVs" curve represents the actual power generation of the genset in each period when PEVs are engaged. The curve "Original Demand + FEVs" reduces the grid fluctuation compared with the curve "Original Demand" and achieves the effect of peak and valley reduction. Each stacked histogram in Figure 7 represents the sum of the

power generation of each test case at each period, and the bars with different colors represent the power output of different units. Meanwhile, for **Scenario 2**, the output powers of each unit and PEVs in each period are also shown in Tables 8, A3 and A4, in Appendix A. Table 7 shows that the grid fluctuations for **5 units**, **10 units**, and **15 units** can be reduced by **25.5%**, **48.8%**, and **44.5%**, respectively. Furthermore, for **5 units**, **10 units**, and **15 units**, the fuel cost is also reduced by **1.2%**, **4.8%**, and **3.1%**, respectively. Additionally, for **5 units**, **10 units**, and **15 units**, the CPU running times decreased by **1.2%**, **4.8%**, and **3.1%**, respectively. Therefore, the connection of the electric vehicle to the grid does improve the operation of the power grid, and it further proves that the CMFO is very effective for solving DED with PEVs [49,50].

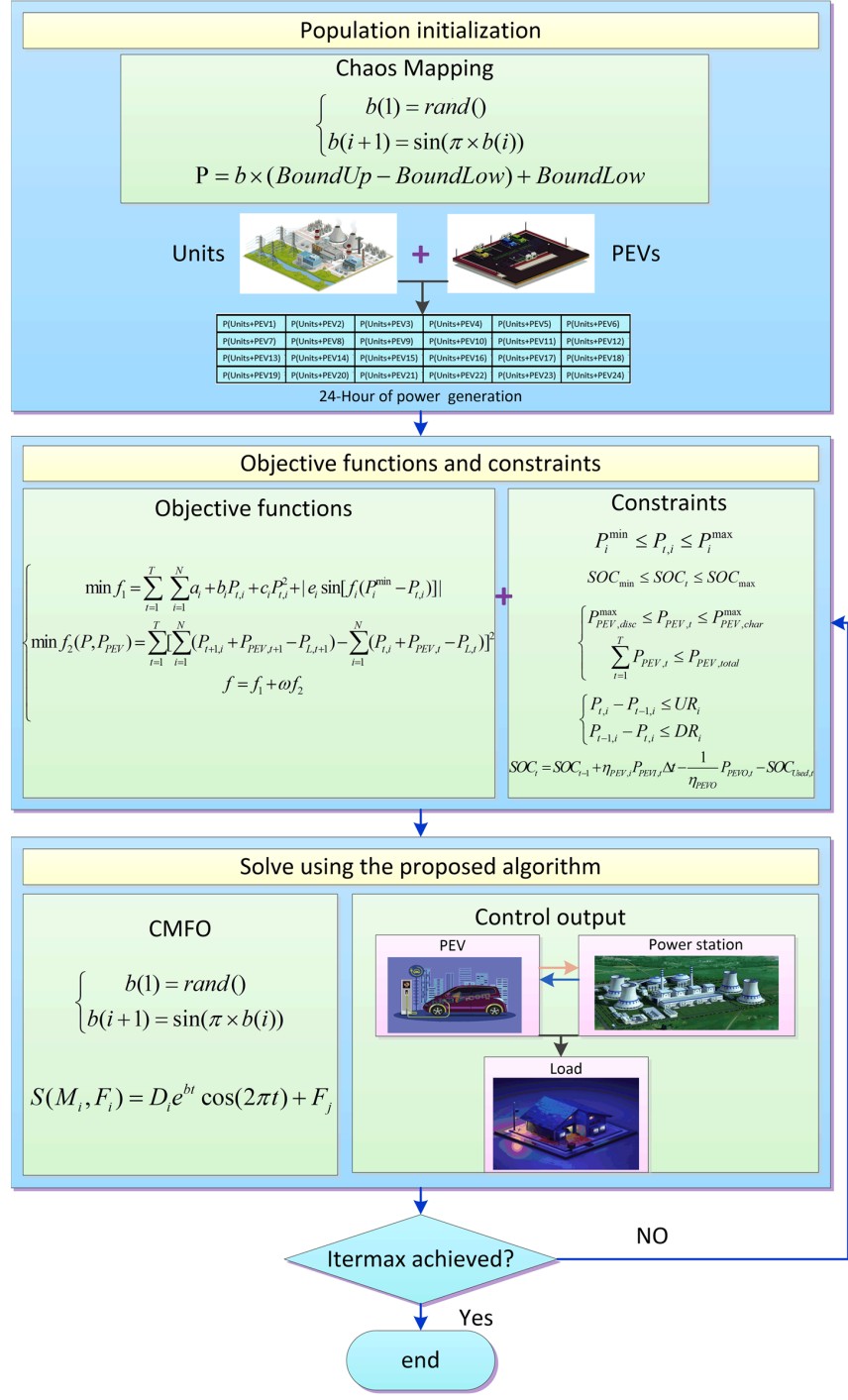

**Figure 4.** Overall framework of the CMFO for DED.

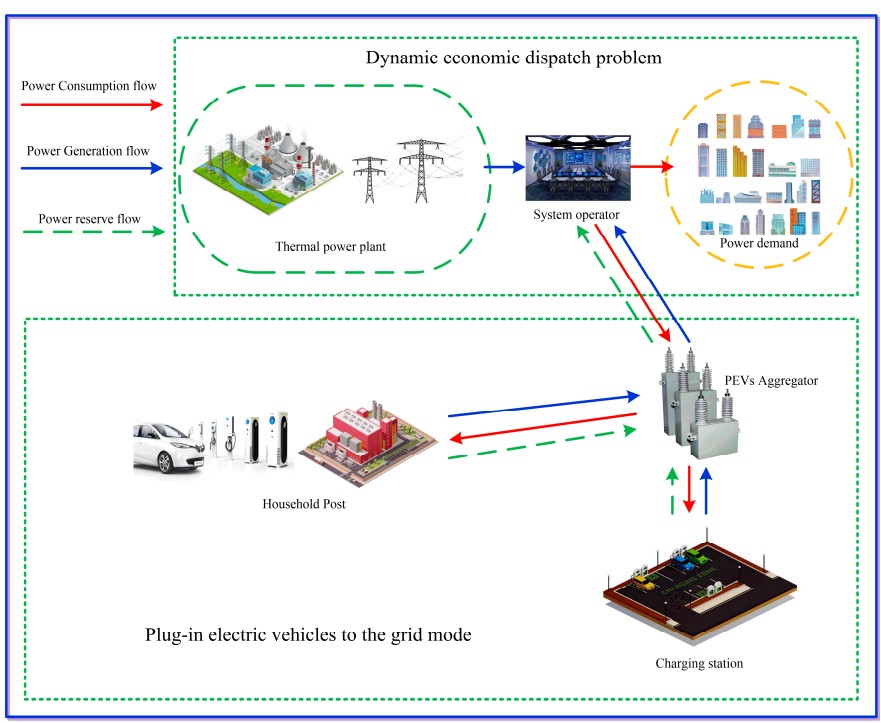

**Figure 5.** The model of plug-in electric vehicles to the grid.

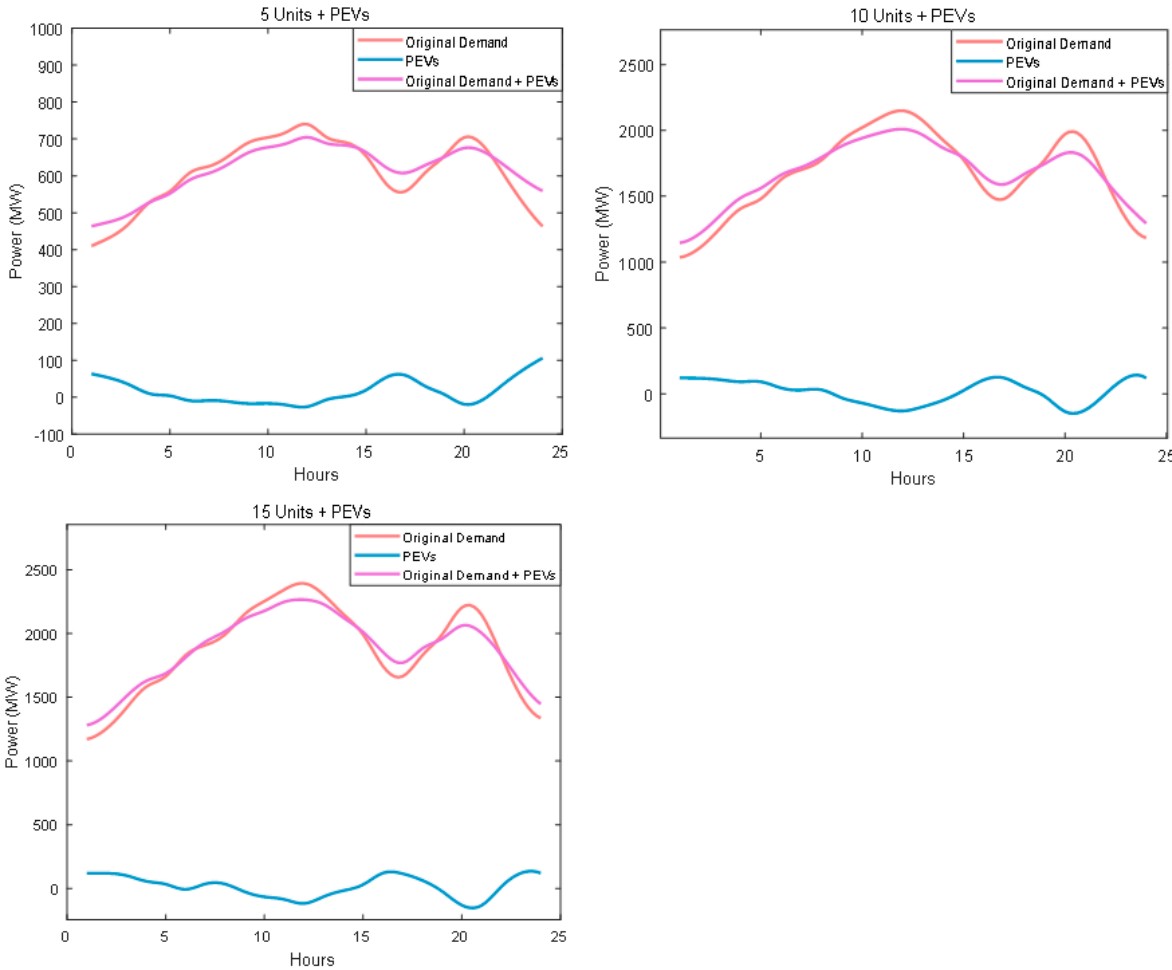

**Figure 6.** Output power of Units + PEVs.

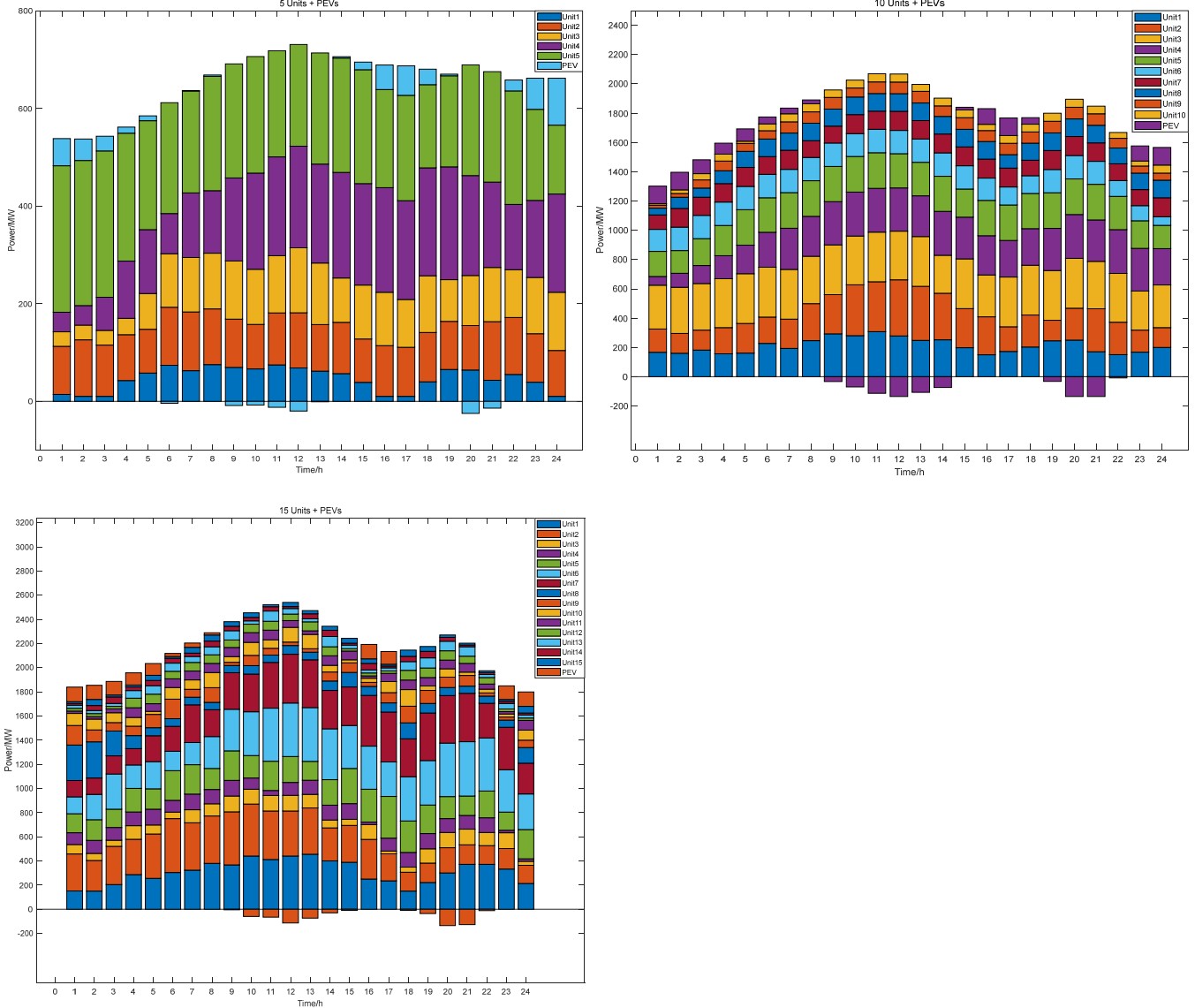

**Figure 7.** Power curve of Units + PEVs.

### *5.3. Performance Evaluation of CMFO*

To test the performance of the CMFO for DED, the CMFO is compared with GADMFI [26], MFO [27], and MPA [35]. The experiment results are shown in Table 9, of which **'NFS'** is represented as an infeasible solution. From Table 9, compared with GADMFI [26] in the six cases, the CMFO reduces the best fuel costs by **7.4%, 7.3%, 0.2%, 10.5%, 12.5%**, and **1.1%**; reduces the worst fuel costs by **6.8%, 6.7%, −2.7%, 4.8%, 6.4%**, and **1.3%**; reduces the average fuel costs by **8.0%, 7.6%, 0.2%, 5.4%, 10.1%**, and **−1.2%**, respectively; and reduces CPU running time by **44.4%, 33.8%, 33.9%, 58.5%, 54.5%**, and **60.0%**, respectively. Compared with MFO [27], the CMFO reduces the best fuel cost by **1.5%, 0.4%, 0%, 2.7%, 0.5%**, and **2.1%**, and reduces CPU running time by **−8.0%, −6.9%, 2.8%, 12.1%, 1.8%**, and **1.5%**, respectively. Compared with MPA [35], the CMFO reduces the best fuel cost by **5.0%, 1.1%, 0.1%, 9.9%, 7.4%**, and **2.4%**; reduces the worst fuel costs by **7.9%, 0.3%, 1.9%, 4.8%, 3.0%**, and **1.0%**; reduces the average fuel costs by **5.5%, 1.0%, 1.1%, 7.2%, 4.9%**, and **1.3%**; and reduces CPU running time by **27.2%, 31.1%, 13.5%, 50.6%, 36.7%**, and **16.3%**, respectively. Such improvements mainly benefit from introducing the chaotic mapping mechanism to CMFO, which helps the CMFO to jump out of the local optimum effectively.

**Table 8.** Case IV: 5 units + PEVs.

| Hour | U1(MW) | U2(MW) | U3(MW) | U4(MW) | U5(MW) | PEVs(MW) |
|------|--------|--------|--------|--------|--------|----------|
| 1 | 10.0000 | 53.6858 | 175.0000 | 40.0000 | 50.0000 | 63.9108 |
| 2 | 15.7117 | 27.1060 | 31.5891 | 84.2206 | 93.7074 | 52.5612 |
| 3 | 12.3513 | 62.5644 | 32.9835 | 151.6454 | 295.3407 | 35.8271 |
| 4 | 66.4852 | 74.7164 | 45.3262 | 191.0710 | 125.7262 | 17.3350 |
| 5 | 10.3371 | 47.6915 | 149.1630 | 40.0000 | 90.4991 | 13.1349 |
| 6 | 70.0996 | 83.7379 | 102.7415 | 40.0000 | 50.0000 | 0.6600 |
| 7 | 57.0166 | 42.8705 | 30.0000 | 76.7857 | 112.7795 | 6.9656 |
| 8 | 10.0000 | 52.0368 | 30.0000 | 132.5563 | 50.0000 | 3.7592 |
| 9 | 45.5688 | 98.9987 | 30.0000 | 58.6576 | 286.4174 | −3.9636 |
| 10 | 51.6049 | 43.4705 | 175.0000 | 143.2037 | 50.0000 | −4.3255 |
| 11 | 10.0000 | 20.0000 | 30.0000 | 40.0000 | 155.1979 | −9.3044 |
| 12 | 42.5523 | 97.3219 | 141.7528 | 172.2989 | 63.4971 | −20.6566 |
| 13 | 10.0000 | 44.5992 | 30.0000 | 250.0000 | 150.2540 | −3.7945 |
| 14 | 10.0000 | 118.0385 | 157.8564 | 244.6124 | 50.0000 | 2.2380 |
| 15 | 19.3010 | 33.3553 | 133.4780 | 99.8295 | 50.0000 | 15.9469 |
| 16 | 27.0477 | 20.0000 | 100.0908 | 111.1224 | 202.6196 | 49.0711 |
| 17 | 75.0000 | 124.5431 | 93.7788 | 40.0000 | 137.5331 | 55.8396 |
| 18 | 59.3929 | 68.8096 | 169.7482 | 102.9944 | 50.0000 | 26.3521 |
| 19 | 75.0000 | 53.0772 | 83.2589 | 40.0000 | 271.9666 | −1.6668 |
| 20 | 10.0000 | 85.7967 | 108.6016 | 40.0000 | 101.0388 | −31.0459 |
| 21 | 29.7327 | 39.2235 | 46.4968 | 40.0000 | 174.5293 | −20.2485 |
| 22 | 47.7149 | 106.2207 | 153.9936 | 100.3830 | 300.0000 | 18.7699 |
| 23 | 27.1922 | 84.3751 | 30.0000 | 59.3461 | 131.1061 | 56.5571 |
| 24 | 75.0000 | 104.5927 | 39.5150 | 152.5159 | 50.0000 | 87.0775 |
| **Total fuel cost ($): 3.93 $\times$ 10$^4$** | | | | | | |

**Table 9.** Comparison of different algorithms for DED.

| Test Case | Algorithm | Fuel Costs ($) | | | CPU Time (s) |
|-----------|-----------|----------------|----------|----------|--------------|
| | | **Best ($)** | **Worst ($)** | **Average ($)** | **Average (s)** |
| **Case I** | GADMFI | $4.3085 \times 10^4$ | $4.3145 \times 10^4$ | $4.3109 \times 10^4$ | $4.77 \times 10^2$ |
| | MFO | $4.04 \times 10^4$ | NFS | NFS | $2.45 \times 10^2$ |
| | MPA | $4.18 \times 10^4$ | $4.327 \times 10^4$ | $4.217 \times 10^4$ | $3.37 \times 10^2$ |
| | CMFO | **$3.98 \times 10^4$** | **$4.07 \times 10^4$** | **$3.99 \times 10^4$** | **$2.65 \times 10^2$** |
| **Case II** | GADMFI | $2.4643 \times 10^6$ | $2.4649 \times 10^6$ | $2.4646 \times 10^6$ | $1.4344 \times 10^3$ |
| | MFO | $2.29 \times 10^6$ | $3.29 \times 10^6$ | $3.11 \times 10^6$ | $8.95 \times 10^2$ |
| | MPA | $2.3047 \times 10^6$ | $2.3162 \times 10^6$ | $2.3119 \times 10^6$ | $1.2549 \times 10^3$ |
| | CMFO | **$2.28 \times 10^6$** | **$2.31 \times 10^6$** | **$2.29 \times 10^6$** | **$9.57 \times 10^2$** |
| **Case III** | GADMFI | $6.7313 \times 10^5$ | $6.7690 \times 10^5$ | $6.7561 \times 10^5$ | $3.1073 \times 10^3$ |
| | MFO | $6.71 \times 10^5$ | NFS | NFS | $2.11 \times 10^3$ |
| | MPA | $6.715 \times 10^5$ | $7.082 \times 10^5$ | $6.814 \times 10^5$ | $2.3261 \times 10^3$ |
| | CMFO | **$6.71 \times 10^5$** | **$6.95 \times 10^5$** | **$6.74 \times 10^5$** | **$2.05 \times 10^3$** |
| **Case IV** | GADMFI | $4.3944 \times 10^4$ | $4.3900 \times 10^4$ | $4.3897 \times 10^4$ | $5.59 \times 10^2$ |
| | MFO | $4.04 \times 10^4$ | NFS | NFS | $2.65 \times 10^2$ |
| | MPA | $4.320 \times 10^4$ | $4.393 \times 10^4$ | $4.3644 \times 10^4$ | $3.51 \times 10^2$ |
| | CMFO | **$3.93 \times 10^4$** | **$4.19 \times 10^4$** | **$4.07 \times 10^4$** | **$2.33 \times 10^2$** |
| **Case V** | GADMFI | $2.4818 \times 10^6$ | $2.4801 \times 10^6$ | $2.4811 \times 10^6$ | $2.0429 \times 10^3$ |
| | MFO | $2.18 \times 10^6$ | $3.29 \times 10^6$ | $3.09 \times 10^6$ | $9.48 \times 10^2$ |
| | MPA | $2.3304 \times 10^6$ | $2.3992 \times 10^6$ | $2.3594 \times 10^6$ | $1.2709 \times 10^3$ |
| | CMFO | **$2.17 \times 10^6$** | **$2.33 \times 10^6$** | **$2.25 \times 10^6$** | **$9.30 \times 10^2$** |
| **Case VI** | GADMFI | $6.5694 \times 10^5$ | $6.7391 \times 10^5$ | $6.6320 \times 10^5$ | $4.88 \times 10^3$ |
| | MFO | $6.64 \times 10^5$ | NFS | NFS | $2.00 \times 10^3$ |
| | MPA | $6.658 \times 10^5$ | $6.882 \times 10^5$ | $6.798 \times 10^5$ | $2.2902 \times 10^3$ |
| | CMFO | **$6.50 \times 10^5$** | **$6.83 \times 10^5$** | **$6.71 \times 10^5$** | **$1.97 \times 10^3$** |



## 6. Conclusions

This paper explores a modified version of the moth flame algorithm (MFO), referred to as CMFO, to solve the DED problem in the presence of PEVs in a more efficient manner. The proposed algorithm incorporates chaotic mapping to initialize the population and improve the evolution mechanism of MFO, thereby compensating for its shortcomings and avoiding premature convergence. The performance of the CMFO is evaluated through experiments on benchmark functions and compared with other popular algorithms. The results demonstrate that CMFO can find high-quality solutions in a shorter time frame. Furthermore, two scenarios with and without PEVs are tested using CMFO, with the number of units ranging from 5 to 15. The results show that CMFO can produce optimal solutions, especially for large-scale problems and confirm that V2G can effectively reduce grid fluctuations and achieve peak shaving and valley filling. In the future, we will consider the introduction of solar energy as well as wind energy into the DED problem, which will be more helpful to solve the actual problem.

**Author Contributions:** Conceptualization, W.Y., X.Z. and F.N.; Data curation, Z.Y.; Formal analysis, W.Y. and X.Z.; Funding acquisition, W.Y., H.J. and Z.Y.; Methodology, W.Y. and X.Z.; Project administration, W.Y.; Resources, W.Y. and X.Z.; Software, X.Z.; Supervision, Q.X.; Visualization, Z.Y.; Writing—review & editing, W.Y., X.Z., F.N., H.J., Q.X. and Z.Y. All authors have read and agreed to the published version of the manuscript.

**Funding:** This research work is supported by the National Natural Science Foundation of China (Nos. 11871196 and 12071133), the National Key Research and Development Project of China (2018YFB1700500), the Scientific and Technological Project of Henan Province (No. 222102110095), and the Higher Learning Key Development Project of Henan Province (No. 22A120007), the Natural Science Foundation of Guangdong Province (2020A1515110541).

**Informed Consent Statement:** Not applicable.

**Data Availability Statement:** Data is contained within the article.

**Acknowledgments:** This research work is supported by the National Natural Science Foundation of China (Nos. 11871196 and 12071133), the National Key Research and Development Project of China (2018YFB1700500), the Scientific and Technological Project of Henan Province (No. 222102110095), and the Higher Learning Key Development Project of Henan Province (No. 22A120007), the Natural Science Foundation of Guangdong Province (2020A1515110541).

**Conflicts of Interest:** The authors declare no conflict of interest.

## Appendix A

**Table A1.** Comparison of CMFO with other algorithms (20 dimensions).

| NO. | Statistics | ALO | DA | GOA | MVO | SSA | WOA | SCA | MPA | GWO | MFO | CMFO |
|---|---|---|---|---|---|---|---|---|---|---|---|---|
| **F1** | **Sta** | $2.12 \times 10^{-8}$ | $3.83 \times 10^2$ | $3.51 \times 10^{-2}$ | $1.67 \times 10^{-2}$ | $5.81 \times 10^{-10}$ | $6.02 \times 10^{-154}$ | $4.47 \times 10^{-8}$ | $1.7 \times 10^{-52}$ | $5.18 \times 10^{-75}$ | $6.85 \times 10^{-8}$ | $1.12 \times 10^{-11}$ |
| | **Best** | $2.17 \times 10^{-8}$ | $1.22 \times 10^2$ | $6.54 \times 10^{-4}$ | $4.64 \times 10^{-2}$ | $2.87 \times 10^{-9}$ | $7.49 \times 10^{-167}$ | $5.93 \times 10^{-11}$ | $1.0 \times 10^{-57}$ | $2.39 \times 10^{-78}$ | $3.05 \times 10^{-6}$ | $1.80 \times 10^{-13}$ |
| | **Time(s)** | $3.65 \times 10^1$ | $6.78 \times 10^1$ | $1.16 \times 10^2$ | $1.29 \times 10^0$ | $4.39 \times 10^{-1}$ | $1.13 \times 10^0$ | $6.39 \times 10^{-1}$ | $1.03 \times 10^0$ | $9.41 \times 10^{-1}$ | $1.39 \times 10^0$ | $1.15 \times 10^0$ |
| | **Winner** | N | N | N | N | N | Y | N | Y | Y | N | + |
| **F2** | **Sta** | $1.93 \times 10^1$ | $3.09 \times 10^0$ | $2.95 \times 10^{15}$ | $1.52 \times 10^{-2}$ | $4.76 \times 10^{-1}$ | $9.18 \times 10^{-102}$ | $3.51 \times 10^{-9}$ | $5.9 \times 10^{-29}$ | $1.13 \times 10^{-43}$ | $1.72 \times 10^{-6}$ | $4.07 \times 10^{-9}$ |
| | **Best** | $1.13 \times 10^{-2}$ | $3.41 \times 10^0$ | $2.51 \times 10^2$ | $7.71 \times 10^{-2}$ | $5.53 \times 10^{-5}$ | $1.71 \times 10^{-113}$ | $2.11 \times 10^{-11}$ | $1.4 \times 10^{-32}$ | $4.94 \times 10^{-45}$ | $8.00 \times 10^{-5}$ | $8.16 \times 10^{-10}$ |
| | **Time(s)** | $4.10 \times 10^1$ | $8.22 \times 10^1$ | $1.29 \times 10^2$ | $1.38 \times 10^0$ | $5.00 \times 10^{-1}$ | $1.22 \times 10^0$ | $6.54 \times 10^{-1}$ | $1.23 \times 10^0$ | $1.05 \times 10^0$ | $10.31 \times 10^{-1}$ | $9.85 \times 10^{-1}$ |
| | **Winner** | N | N | N | N | N | Y | Y | Y | Y | N | + |
| **F3** | **Sta** | $5.36 \times 10^1$ | $2.53 \times 10^3$ | $8.77 \times 10^1$ | $1.32 \times 10^0$ | $7.71 \times 10^{-1}$ | $1.93 \times 10^3$ | $2.81 \times 10^2$ | $5.0 \times 10^{-18}$ | $2.66 \times 10^{-23}$ | 11.8826 | $6.59 \times 10^0$ |
| | **Best** | $8.58 \times 10^0$ | $4.35 \times 10^2$ | $3.15 \times 10^1$ | $6.35 \times 10^{-1}$ | $2.62 \times 10^{-2}$ | $2.72 \times 10^2$ | $1.92 \times 10^{-1}$ | $4.5 \times 10^{-27}$ | $2.51 \times 10^{-30}$ | $3.99 \times 10^3$ | $7.84 \times 10^{-1}$ |
| | **Time(s)** | $4.27 \times 10^1$ | $9.03 \times 10^1$ | $1.29 \times 10^2$ | $1.65 \times 10^0$ | $8.20 \times 10^{-1}$ | $1.76 \times 10^0$ | $9.26 \times 10^{-1}$ | $1.95 \times 10^0$ | $1.36 \times 10^0$ | $1.57 \times 10^0$ | $1.52 \times 10^0$ |
| | **Winner** | N | N | N | N | N | N | N | Y | Y | N | + |
| **F4** | **Sta** | $2.83 \times 10^{-8}$ | $1.41 \times 10^0$ | $3.51 \times 10^{-2}$ | $2.19 \times 10^{-2}$ | $1.35 \times 10^{-9}$ | $1.56 \times 10^{-2}$ | $1.71 \times 10^{-1}$ | $7.40 \times 10^{-11}$ | $1.23 \times 10^{-1}$ | $4.02 \times 10^{-4}$ | $3.24 \times 10^{-11}$ |
| | **Best** | $2.35 \times 10^{-8}$ | $3.18 \times 10^{-1}$ | $8.17 \times 10^{-4}$ | $2.75 \times 10^{-2}$ | $1.82 \times 10^{-9}$ | $7.99 \times 10^{-4}$ | $1.68 \times 10^0$ | $4.58 \times 10^{-11}$ | $2.21 \times 10^{-6}$ | $5.13 \times 10^{-8}$ | $8.78 \times 10^{-13}$ |
| | **Time(s)** | $4.19 \times 10^1$ | $6.20 \times 10^1$ | $1.55 \times 10^2$ | $1.47 \times 10^0$ | $5.09 \times 10^{-1}$ | $1.17 \times 10^0$ | $6.22 \times 10^{-1}$ | $1.17 \times 10^0$ | $1.03 \times 10^0$ | $9.99 \times 10^{-1}$ | $9.72 \times 10^{-1}$ |
| | **Winner** | N | N | N | N | N | N | N | N | N | N | + |
| **F5** | **Sta** | $2.85 \times 10^{-2}$ | $9.47 \times 10^{-1}$ | $9.49 \times 10^{-2}$ | $7.82 \times 10^{-2}$ | $1.31 \times 10^{-2}$ | $0.00 \times 10^0$ | $2.32 \times 10^{-1}$ | $0.00 \times 10^0$ | $0.00 \times 10^0$ | 52.8037 | $1.60 \times 10^{-12}$ |
| | **Best** | $5.38 \times 10^{-6}$ | $8.84 \times 10^{-1}$ | $3.78 \times 10^{-2}$ | $1.29 \times 10^{-1}$ | $1.34 \times 10^{-8}$ | $0.00 \times 10^0$ | $1.63 \times 10^{-9}$ | $0.00 \times 10^0$ | $0.00 \times 10^0$ | 40.6902 | $4.66 \times 10^{-13}$ |
| | **Time(s)** | $4.23 \times 10^1$ | $6.89 \times 10^1$ | $1.91 \times 10^2$ | $1.54 \times 10^0$ | $5.85 \times 10^{-1}$ | $1.31 \times 10^0$ | $7.00 \times 10^{-1}$ | $1.28 \times 10^0$ | $1.11 \times 10^0$ | $1.20 \times 10^0$ | $1.09 \times 10^0$ |
| | **Winner** | N | N | N | N | N | Y | N | Y | Y | N | + |
| **F6** | **Sta** | $2.62 \times 10^0$ | $5.89 \times 10^0$ | $1.96 \times 10^0$ | $4.90 \times 10^{-1}$ | $9.69 \times 10^{-1}$ | $2.54 \times 10^{-3}$ | $1.17 \times 10^{-1}$ | $1.31 \times 10^{-11}$ | $1.53 \times 10^{-2}$ | 3.0935 | $3.10 \times 10^{-10}$ |
| | **Best** | $4.37 \times 10^{-1}$ | $7.08 \times 10^{-1}$ | $1.04 \times 10^0$ | $6.52 \times 10^{-4}$ | $3.30 \times 10^{-1}$ | $3.81 \times 10^{-4}$ | $2.50 \times 10^{-1}$ | $5.06 \times 10^{-12}$ | $9.72 \times 10^{-3}$ | $6.47 \times 10^{-5}$ | $1.77 \times 10^{-12}$ |
| | **Time(s)** | $4.29 \times 10^1$ | $7.69 \times 10^1$ | $2.01 \times 10^2$ | $1.93 \times 10^0$ | $1.07 \times 10^0$ | $2.21 \times 10^0$ | $1.16 \times 10^0$ | $2.30 \times 10^0$ | $1.58 \times 10^0$ | $1.79 \times 10^0$ | $1.82 \times 10^0$ |
| | **Winner** | N | N | N | N | N | N | N | N | N | N | + |
| **F7** | **Sta** | $7.16 \times 10^{-3}$ | $8.54 \times 10^0$ | $8.36 \times 10^{-2}$ | $1.65 \times 10^{-2}$ | $4.39 \times 10^{-3}$ | $8.02 \times 10^{-2}$ | $6.16 \times 10^{-2}$ | $8.67 \times 10^{-11}$ | $1.26 \times 10^{-1}$ | $7.86 \times 10^{-4}$ | $5.12 \times 10^{-11}$ |
| | **Best** | $3.96 \times 10^{-8}$ | $1.49 \times 10^0$ | $4.09 \times 10^{-3}$ | $5.97 \times 10^{-3}$ | $1.75 \times 10^{-10}$ | $7.50 \times 10^{-3}$ | $1.17 \times 10^0$ | $4.80 \times 10^{-11}$ | $1.37 \times 10^{-5}$ | $3.29 \times 10^{-7}$ | $1.34 \times 10^{-11}$ |
| | **Time(s)** | $4.27 \times 10^1$ | $7.68 \times 10^1$ | $2.65 \times 10^2$ | $1.97 \times 10^0$ | $1.07 \times 10^0$ | $2.23 \times 10^0$ | $1.18 \times 10^0$ | $2.32 \times 10^0$ | $1.70 \times 10^0$ | $1.86 \times 10^0$ | $1.80 \times 10^0$ |
| | **Winner** | N | N | N | N | N | N | N | N | N | N | + |

**Table A2.** Comparison of CMFO with other algorithms (50 dimensions).

| NO. | Statistics | ALO | DA | GOA | MVO | SSA | WOA | SCA | MPA | GWO | MFO | CMFO |
|---|---|---|---|---|---|---|---|---|---|---|---|---|
| F1 | Sta | $2.96 \times 10^{-27}$ | $1.92 \times 10^{2}$ | $7.30 \times 10^{-3}$ | $7.90 \times 10^{-3}$ | $1.93 \times 10^{-9}$ | $0.00 \times 10^{0}$ | $1.35 \times 10^{-16}$ | $0.00 \times 10^{0}$ | $0.00 \times 10^{0}$ | $5.35 \times 10^{-4}$ | $5.71 \times 10^{-38}$ |
| | Best | $8.27 \times 10^{-26}$ | $5.07 \times 10^{1}$ | $1.66 \times 10^{-2}$ | $1.47 \times 10^{-2}$ | $9.04 \times 10^{-9}$ | $0.00 \times 10^{0}$ | $1.01 \times 10^{-28}$ | $0.00 \times 10^{0}$ | $0.00 \times 10^{0}$ | 0.0633 | $2.95 \times 10^{-39}$ |
| | Time(s) | $3.71 \times 10^{3}$ | $7.07 \times 10^{2}$ | $3.13 \times 10^{3}$ | $3.18 \times 10^{1}$ | $7.46 \times 10^{0}$ | $1.73 \times 10^{1}$ | $1.20 \times 10^{1}$ | $2.46 \times 10^{1}$ | 22.47876667 | 9.991 | 9.828 |
| | Winner | N | N | N | N | N | Y | N | Y | Y | N | + |
| F2 | Sta | $3.5048 \times 10^{-6}$ | $2.99 \times 10^{0}$ | $5.41 \times 10^{23}$ | $5.69 \times 10^{-2}$ | $1.42 \times 10^{0}$ | $0.00 \times 10^{0}$ | $7.51 \times 10^{-29}$ | $0.00 \times 10^{0}$ | $0.00 \times 10^{0}$ | 10 | $2.87 \times 10^{1}$ |
| | Best | $1.40 \times 10^{-7}$ | $1.75 \times 10^{0}$ | $5.16 \times 10^{22}$ | $1.09 \times 10^{-1}$ | $4.20 \times 10^{-3}$ | $0.00 \times 10^{0}$ | $4.22 \times 10^{-42}$ | $5.92 \times 10^{-286}$ | $3.21 \times 10^{-277}$ | 42.4382 | $3.31 \times 10^{-22}$ |
| | Time(s) | $3.61 \times 10^{3}$ | $7.47 \times 10^{2}$ | $4.46 \times 10^{3}$ | $2.91 \times 10^{1}$ | $7.50 \times 10^{0}$ | $1.71 \times 10^{1}$ | $1.14 \times 10^{1}$ | $2.37 \times 10^{1}$ | $2.25 \times 10^{1}$ | $1.11 \times 10^{1}$ | $1.07 \times 10^{1}$ |
| | Winner | N | N | N | N | N | Y | Y | Y | Y | N | + |
| F3 | Sta | $7.83 \times 10^{-1}$ | $8.91 \times 10^{3}$ | $6.58 \times 10^{1}$ | $6.90 \times 10^{-3}$ | $1.43 \times 10^{-4}$ | $3.53 \times 10^{3}$ | $7.62 \times 10^{3}$ | $6.20 \times 10^{-118}$ | $3.02 \times 10^{-103}$ | $2.50 \times 10^{4}$ | $1.95 \times 10^{4}$ |
| | Best | $4.02 \times 10^{0}$ | $7.91 \times 10^{3}$ | $7.46 \times 10^{2}$ | $9.97 \times 10^{0}$ | $1.58 \times 10^{-5}$ | $2.75 \times 10^{0}$ | $5.69 \times 10^{2}$ | $9.46 \times 10^{-170}$ | $7.69 \times 10^{-125}$ | $1.69 \times 10^{4}$ | $3.83 \times 10^{-1}$ |
| | Time(s) | $4.07 \times 10^{4}$ | $9.24 \times 10^{2}$ | $3.25 \times 10^{3}$ | $3.67 \times 10^{1}$ | $1.62 \times 10^{1}$ | $3.19 \times 10^{1}$ | $1.91 \times 10^{1}$ | $5.05 \times 10^{1}$ | $3.18 \times 10^{1}$ | $1.81 \times 10^{1}$ | $1.73 \times 10^{1}$ |
| | Winner | N | N | N | N | Y | N | Y | Y | Y | N | + |
| F4 | Sta | $5.88 \times 10^{-9}$ | $1.71 \times 10^{2}$ | $1.58 \times 10^{-1}$ | $2.11 \times 10^{-2}$ | $2.11 \times 10^{-9}$ | $2.77 \times 10^{-5}$ | $3.65 \times 10^{-1}$ | $1.50 \times 10^{-12}$ | $7.87 \times 10^{-1}$ | $8.94 \times 10^{-2}$ | $2.62 \times 10^{-28}$ |
| | Best | $8.62 \times 10^{-8}$ | $5.19 \times 10^{2}$ | $2.37658 \times 10^{-6}$ | $1.61 \times 10^{-2}$ | $8.49 \times 10^{-9}$ | $4.51 \times 10^{-5}$ | $7.63 \times 10^{0}$ | $3.38 \times 10^{-12}$ | $7.42 \times 10^{-1}$ | $9.15 \times 10^{-7}$ | $3.89 \times 10^{-30}$ |
| | Time(s) | $3.95 \times 10^{3}$ | $6.28 \times 10^{2}$ | $3.28 \times 10^{3}$ | $3.07 \times 10^{1}$ | $7.52 \times 10^{0}$ | $2.13 \times 10^{1}$ | $1.12 \times 10^{1}$ | $2.36 \times 10^{1}$ | $2.24 \times 10^{1}$ | $10.06 \times 10^{0}$ | $9.54 \times 10^{0}$ |
| | Winner | N | N | N | N | N | N | N | N | N | N | N |
| F5 | Sta | $5.80 \times 10^{-5}$ | $1.05 \times 10^{0}$ | $9.34 \times 10^{-8}$ | $6.50 \times 10^{-2}$ | $7.20 \times 10^{-3}$ | $3.00 \times 10^{-3}$ | $1.17 \times 10^{-1}$ | $0.00 \times 10^{0}$ | $0.00 \times 10^{0}$ | 219.95 | $0.00 \times 10^{0}$ |
| | Best | $1.51 \times 10^{-8}$ | $1.79 \times 10^{0}$ | $1.0899 \times 10^{-8}$ | $5.51 \times 10^{-2}$ | $1.76 \times 10^{-8}$ | $0.00 \times 10^{0}$ | $0.00 \times 10^{0}$ | $0.00 \times 10^{0}$ | $0.00 \times 10^{0}$ | 40.93 | $0.00 \times 10^{0}$ |
| | Time(s) | $3.99 \times 10^{3}$ | $6.96 \times 10^{2}$ | $3.29 \times 10^{3}$ | $3.21 \times 10^{1}$ | $8.67 \times 10^{0}$ | $2.03 \times 10^{1}$ | $1.21 \times 10^{1}$ | $2.60 \times 10^{1}$ | $2.31 \times 10^{1}$ | $1.20 \times 10^{1}$ | $1.15 \times 10^{1}$ |
| | Winner | N | N | N | N | N | N | N | N | N | N | + |
| F6 | Sta | $2.66 \times 10^{0}$ | $1.84 \times 10^{0}$ | $1.6563 \times 10^{-7}$ | $1.04 \times 10^{0}$ | $2.13 \times 10^{0}$ | $2.15 \times 10^{-6}$ | $4.02 \times 10^{-1}$ | $5.11 \times 10^{-14}$ | $2.09 \times 10^{-2}$ | 9.55 | $2.69 \times 10^{-28}$ |
| | Best | $8.02 \times 10^{0}$ | $4.59 \times 10^{0}$ | $1.10373 \times 10^{-8}$ | $1.01 \times 10^{-4}$ | $8.16 \times 10^{-2}$ | $2.84 \times 10^{-6}$ | $4.98 \times 10^{-1}$ | $1.00 \times 10^{-14}$ | $4.33 \times 10^{-2}$ | $8.03 \times 10^{-9}$ | $5.01 \times 10^{-32}$ |
| | Time(s) | $4.09 \times 10^{4}$ | $8.24 \times 10^{2}$ | $3.32 \times 10^{3}$ | $3.79 \times 10^{1}$ | $1.72 \times 10^{1}$ | $3.49 \times 10^{1}$ | $2.05 \times 10^{1}$ | $4.67 \times 10^{1}$ | $3.20 \times 10^{1}$ | $1.95 \times 10^{1}$ | $1.90 \times 10^{1}$ |
| | Winner | N | N | N | N | N | N | N | N | N | N | + |
| F7 | Sta | $5.20 \times 10^{-3}$ | $9.55 \times 10^{0}$ | $1.52238 \times 10^{-7}$ | $5.30 \times 10^{-3}$ | $5.40 \times 10^{-3}$ | $4.10 \times 10^{-3}$ | $5.33 \times 10^{0}$ | $3.30 \times 10^{-3}$ | $3.19 \times 10^{-1}$ | $3.75 \times 10^{-2}$ | $3.81 \times 10^{-27}$ |
| | Best | $5.96 \times 10^{-9}$ | $1.65 \times 10^{1}$ | $5.83674 \times 10^{-9}$ | $1.70 \times 10^{-3}$ | $4.06 \times 10^{-10}$ | $6.05 \times 10^{-5}$ | $3.79 \times 10^{0}$ | $2.93 \times 10^{-12}$ | $1.00 \times 10^{0}$ | $1.94 \times 10^{-7}$ | $1.38 \times 10^{-30}$ |
| | Time(s) | $3.91 \times 10^{4}$ | $1.25 \times 10^{3}$ | $4.63 \times 10^{3}$ | $4.00 \times 10^{1}$ | $1.80 \times 10^{1}$ | $3.42 \times 10^{1}$ | $2.06 \times 10^{1}$ | $4.66 \times 10^{1}$ | $3.21 \times 10^{1}$ | $2.04 \times 10^{1}$ | $1.98 \times 10^{1}$ |
| | Winner | N | N | N | N | N | N | N | N | N | N | + |

**Table A3.** Case V: 10 units + PEVs.

| Hour | U1(MW) | U2(MW) | U3(MW) | U4(MW) | U5(MW) | U6(MW) | U7(MW) | U8(MW) | U9(MW) | U10(MW) | PEVs(MW) |
|---|---|---|---|---|---|---|---|---|---|---|---|
| 1 | 160.8696 | 135.0000 | 75.8791 | 70.4196 | 75.0092 | 90.0275 | 62.8406 | 120.0000 | 52.8649 | 10.8690 | 94.6990 |
| 2 | 158.0798 | 209.0647 | 210.1943 | 91.7465 | 110.1556 | 96.0797 | 41.5575 | 120.0000 | 20.0000 | 45.8892 | 95.6250 |
| 3 | 326.0844 | 152.2495 | 340.0000 | 96.2404 | 151.6132 | 106.2034 | 20.0000 | 47.4621 | 52.6828 | 10.0000 | 88.6514 |
| 4 | 218.9264 | 469.0351 | 168.7642 | 287.0613 | 130.2719 | 160.0000 | 113.9288 | 98.3395 | 80.0000 | 32.7801 | 95.6250 |
| 5 | 470.0000 | 135.0000 | 157.5864 | 60.0000 | 73.000 | 122.8604 | 35.9626 | 91.3752 | 21.6378 | 25.7479 | 95.6250 |
| 6 | 265.8216 | 300.9719 | 220.1718 | 60.0000 | 185.2786 | 133.4428 | 94.2156 | 62.6338 | 80.0000 | 10.0000 | 69.1834 |
| 7 | 343.1182 | 429.8119 | 119.9667 | 60.0000 | 132.6423 | 57.0000 | 50.6804 | 116.6768 | 62.8030 | 34.8863 | 69.6282 |
| 8 | 162.6787 | 343.1822 | 340.0000 | 164.6482 | 200.1918 | 92.9345 | 65.3742 | 120.0000 | 74.5129 | 10.0000 | 43.7423 |
| 9 | 207.8677 | 135.0000 | 73.0000 | 60.0000 | 73.0000 | 160.0000 | 104.0212 | 88.8840 | 75.6037 | 21.8873 | −46.6844 |
| 10 | 194.0210 | 265.0607 | 134.8639 | 60.0000 | 80.1453 | 91.1980 | 69.5596 | 91.7487 | 20.0000 | 55.0000 | −67.6524 |
| 11 | 268.7014 | 135.0000 | 222.6136 | 300.0000 | 194.5386 | 160.0000 | 20.0000 | 63.4064 | 73.8344 | 55.0000 | −95.6250 |
| 12 | 208.6174 | 135.0000 | 239.7760 | 60.0000 | 155.4622 | 149.0560 | 100.0996 | 120.0000 | 60.9646 | 54.8210 | −91.0158 |
| 13 | 470.0000 | 251.1638 | 81.6651 | 178.4283 | 188.6079 | 57.0000 | 130.0000 | 105.7340 | 70.6994 | 15.4677 | −89.7206 |
| 14 | 150.0000 | 470.0000 | 73.0000 | 176.1563 | 243.0000 | 143.9798 | 98.1321 | 90.4682 | 76.2892 | 49.0920 | −95.6250 |
| 15 | 227.6801 | 252.8597 | 218.3943 | 296.4656 | 199.4953 | 135.3638 | 103.9355 | 120.0000 | 80.0000 | 12.9426 | −57.0687 |
| 16 | 167.8396 | 135.0000 | 257.8612 | 146.1311 | 111.1340 | 57.0000 | 35.5646 | 59.9987 | 58.8004 | 10.0000 | 95.6250 |
| 17 | 198.1592 | 470.0000 | 172.0344 | 60.0000 | 146.6389 | 57.0000 | 130.0000 | 69.0981 | 67.7218 | 15.6682 | 95.6250 |
| 18 | 152.7733 | 470.0000 | 263.2928 | 277.3604 | 73.0000 | 64.9634 | 121.8852 | 93.0555 | 80.0000 | 10.0000 | 77.2728 |
| 19 | 231.2472 | 135.0000 | 216.8723 | 127.6963 | 169.6135 | 143.6869 | 130.0000 | 49.3555 | 37.4418 | 55.0000 | −21.2253 |
| 20 | 150.0000 | 135.0000 | 340.0000 | 272.0725 | 126.2345 | 160.0000 | 130.0000 | 120.0000 | 57.7948 | 10.0000 | −95.6250 |
| 21 | 360.5431 | 155.0912 | 73.0000 | 300.0000 | 73.0000 | 117.9040 | 124.2923 | 47.0000 | 57.1555 | 30.4884 | −95.6250 |
| 22 | 212.9997 | 201.2310 | 146.0474 | 136.8243 | 243.0000 | 79.0038 | 110.9226 | 120.0000 | 20.0000 | 55.0000 | −2.1954 |
| 23 | 150.0000 | 454.4394 | 168.4946 | 195.1094 | 73.0000 | 160.0000 | 22.4399 | 120.0000 | 24.0680 | 50.3871 | 89.7206 |
| 24 | 202.6265 | 146.0963 | 340.0000 | 300.0000 | 192.4758 | 61.6324 | 27.8024 | 92.8556 | 20.4039 | 55.0000 | 91.5265 |
| Total fuel cost ($): 2.17 × 10$^6$ | | | | | | | | | | | |

**Table A4.** Case VI: 15 units + PEVs.

| Hour | U1(MW) | U2(MW) | U3(MW) | U4(MW) | U5(MW) | U6(MW) | U7(MW) | U8(MW) | U9(MW) | U10(MW) | U11(MW) | U12(MW) | U13(MW) | U14(MW) | U15(MW) | PEVs(MW) |
|------|--------|--------|--------|--------|--------|--------|--------|--------|--------|---------|---------|---------|---------|---------|---------|----------|
| 1 | 176.1425 | 150.0000 | 108.5106 | 37.4253 | 150.0000 | 139.9990 | 149.5835 | 174.3534 | 90.1577 | 145.0685 | 20.0059 | 26.4062 | 32.1544 | 15.6656 | 15.8053 | 95.5845 |
| 2 | 150.0000 | 157.2984 | 76.7167 | 20.9051 | 230.0000 | 135.0000 | 135.0000 | 76.9470 | 25.0000 | 160.0000 | 80.0000 | 44.9757 | 60.3988 | 48.2415 | 55.0000 | 94.9566 |
| 3 | 230.0000 | 189.4391 | 45.0562 | 130.0000 | 257.9667 | 141.3246 | 174.0046 | 71.3305 | 85.0000 | 60.0000 | 62.8225 | 35.5883 | 55.0944 | 47.1280 | 15.0000 | 79.6634 |
| 4 | 250.8802 | 248.6492 | 119.7183 | 46.5227 | 186.8387 | 221.3246 | 254.0046 | 60.0000 | 135.1947 | 92.4861 | 20.0000 | 20.0000 | 66.6012 | 19.8366 | 15.0000 | 16.9080 |
| 5 | 152.0290 | 176.2061 | 130.0000 | 130.0000 | 266.8387 | 301.3246 | 161.6952 | 125.0000 | 162.0000 | 25.0000 | 80.0000 | 49.3656 | 85.0000 | 55.0000 | 29.5266 | 39.0652 |
| 6 | 232.0290 | 256.2061 | 130.0000 | 130.0000 | 346.8387 | 194.7503 | 241.6952 | 92.0639 | 62.0000 | 85.0000 | 80.0000 | 48.3496 | 85.0000 | 55.0000 | 55.0000 | 8.4544 |
| 7 | 185.6283 | 336.2061 | 54.3645 | 130.0000 | 258.0539 | 274.7503 | 321.6952 | 111.4288 | 122.0000 | 145.0000 | 80.0000 | 80.0000 | 71.1359 | 32.9579 | 55.0000 | −20.8501 |
| 8 | 186.1909 | 216.2061 | 123.6913 | 130.0000 | 338.0539 | 274.6801 | 401.6952 | 176.4288 | 100.2891 | 160.0000 | 80.0000 | 80.0000 | 85.0000 | 49.9208 | 24.3312 | 18.4826 |
| 9 | 266.1909 | 267.9316 | 130.0000 | 130.0000 | 218.0539 | 336.0582 | 465.0000 | 151.0490 | 160.2891 | 76.1382 | 80.0000 | 71.6256 | 85.0000 | 55.0000 | 55.0000 | −23.3298 |
| 10 | 237.5543 | 338.4802 | 130.0000 | 130.0000 | 298.0539 | 416.0582 | 465.0000 | 216.0490 | 60.2891 | 133.9989 | 57.8353 | 75.4310 | 76.3099 | 54.4817 | 15.0000 | −62.0132 |
| 11 | 290.1651 | 418.4802 | 130.0000 | 130.0000 | 277.9550 | 296.0582 | 465.0000 | 116.0490 | 120.2891 | 160.0000 | 80.0000 | 80.0000 | 85.0000 | 55.0000 | 55.0000 | −74.9379 |
| 12 | 239.8424 | 455.0000 | 130.0000 | 130.0000 | 357.9550 | 376.0582 | 429.4799 | 181.0490 | 162.0000 | 60.0000 | 80.0000 | 80.0000 | 85.0000 | 55.0000 | 55.0000 | −92.1149 |
| 13 | 314.0770 | 455.0000 | 47.5684 | 35.5678 | 237.9550 | 441.8185 | 383.0594 | 246.0490 | 162.0000 | 120.0000 | 80.0000 | 80.0000 | 85.0000 | 55.0000 | 51.1514 | −62.0901 |
| 14 | 394.0770 | 455.0000 | 74.8172 | 22.3819 | 317.9550 | 321.8185 | 263.0594 | 146.0490 | 62.0000 | 79.2814 | 80.0000 | 57.4652 | 81.4700 | 39.4746 | 46.7670 | 18.2028 |
| 15 | 274.0770 | 445.5679 | 36.2120 | 96.5197 | 245.2786 | 354.0521 | 343.0594 | 61.9443 | 85.9525 | 90.3827 | 25.5028 | 20.0000 | 56.9248 | 55.0000 | 21.6266 | 50.5477 |
| 16 | 154.0770 | 437.4610 | 33.0886 | 54.0824 | 325.2786 | 396.8847 | 223.0594 | 60.0000 | 25.0000 | 143.3122 | 20.0000 | 54.4025 | 25.0000 | 15.0000 | 16.4214 | 94.7860 |
| 17 | 211.8056 | 317.4610 | 69.1444 | 105.9533 | 205.2786 | 320.7043 | 303.0594 | 60.0142 | 25.0000 | 43.3122 | 20.0000 | 52.8721 | 53.8645 | 15.0000 | 55.0000 | 94.7618 |
| 18 | 291.8056 | 207.4020 | 58.8509 | 61.9531 | 241.3096 | 338.9532 | 383.0594 | 125.0142 | 85.0000 | 103.3122 | 76.6928 | 20.0000 | 25.0000 | 55.0000 | 55.0000 | 68.0401 |
| 19 | 371.8056 | 246.0679 | 105.6112 | 127.6078 | 184.5946 | 218.9532 | 463.0594 | 150.3590 | 67.1361 | 78.3030 | 58.3296 | 76.6625 | 53.3198 | 54.6784 | 54.0692 | 3.6578 |
| 20 | 390.0996 | 326.0679 | 130.0000 | 71.8341 | 264.5946 | 298.9532 | 343.0594 | 215.3590 | 117.4928 | 138.3030 | 80.0000 | 80.0000 | 85.0000 | 55.0000 | 55.0000 | −77.6699 |
| 21 | 455.0000 | 406.0679 | 130.0000 | 130.0000 | 200.2990 | 378.9532 | 233.5655 | 280.3590 | 162.0000 | 38.3030 | 47.6016 | 41.6110 | 42.7547 | 15.0000 | 54.7570 | −82.4789 |
| 22 | 368.3029 | 286.0679 | 75.9519 | 95.9541 | 150.0000 | 258.9532 | 313.5655 | 180.3590 | 153.0308 | 65.9215 | 23.2737 | 23.1034 | 85.0000 | 23.4638 | 38.3682 | 32.3405 |
| 23 | 443.9799 | 166.0679 | 32.6274 | 31.8548 | 150.0000 | 162.6467 | 260.4023 | 128.2035 | 53.0308 | 25.0000 | 20.0000 | 76.9612 | 30.3996 | 23.4964 | 18.2893 | 95.4537 |
| 24 | 323.9799 | 194.2043 | 42.2713 | 26.0050 | 150.0000 | 177.8022 | 140.4023 | 88.2984 | 32.0471 | 85.0000 | 80.0000 | 38.9659 | 28.2723 | 55.0000 | 15.0000 | 95.5798 |
| Total fuel cost ($): 649,550 | | | | | | | | | | | | | | | | |

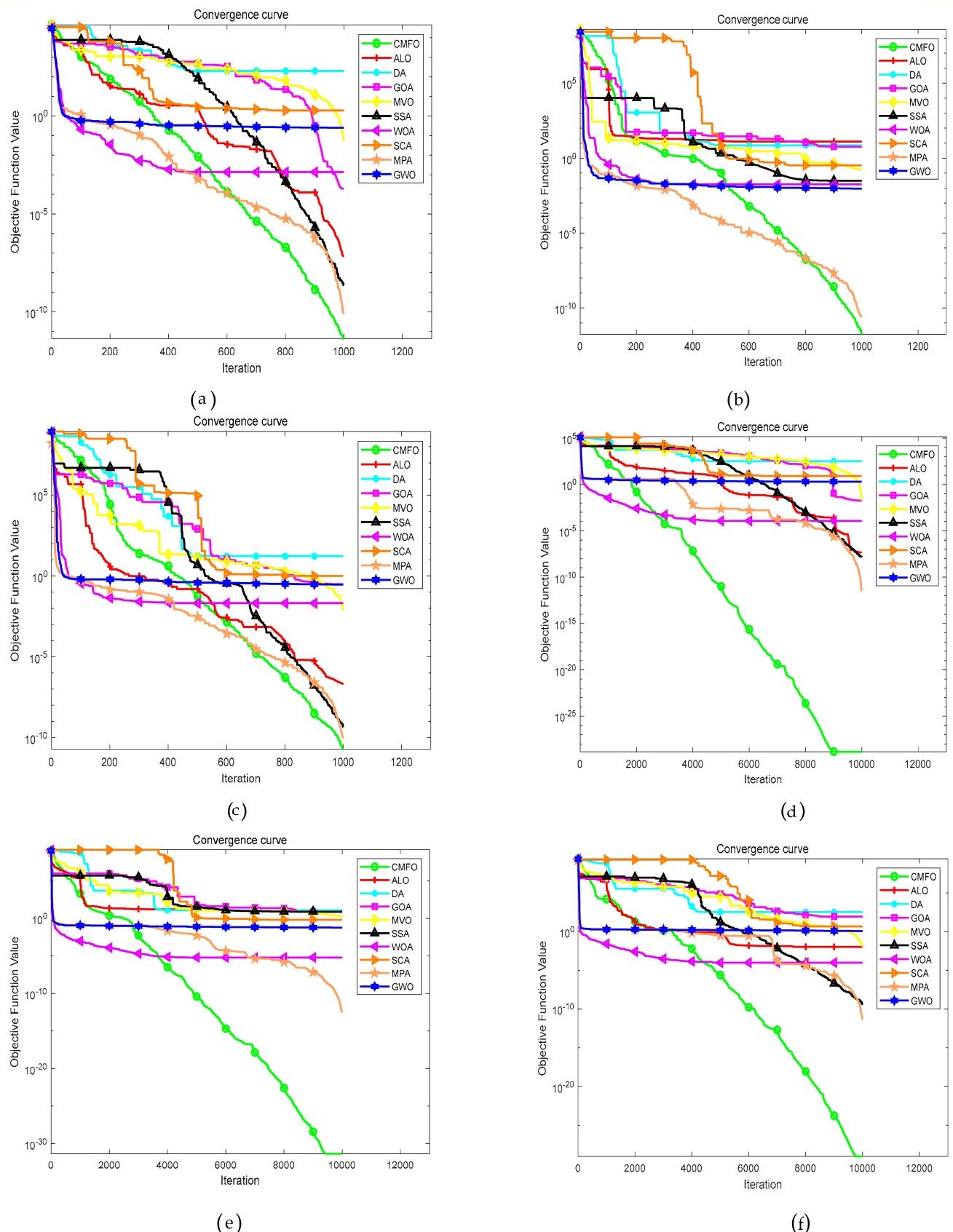

**Figure A1.** The convergence comparison curves of each algorithm. (**a–c**): Test functions are Step, Ackley, and Penalized-1, dimension is 20. (**d–f**): Test functions are Step, Ackley, and Penalized-1, dimension is 50.

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
