# Peer review of "Chaos Moth Flame Algorithm for Multi-Objective Dynamic Economic Dispatch Integrating with Plug-In Electric Vehicles"

_electronics, doi:10.3390/electronics12122742_

Round 1

Reviewer 1 Report

·        The introduction needs a little more structure and flow. I just got trying to understand the purpose of it. Try to be more objective and critical when writing it.

·        I would move the paragraph after table 8 before table 6, it would have been better for understanding.

·        Also, I think the wording of N and Y is a bit confusing can you please be clearer with this?

·        In Tables 6, 7 and 8 why haven’t you compared it to MFO without the chaotic part? It would have been interesting to see the behaviour with and without this new addition. This is a critical question to answer.

·        It is impressive the number of metaheuristic methods that you are comparing to! However, why have you limited that comparison to this method? Why not compare the classical non-heuristic methods such as quadratic programming, gradient descent, and Lagrange relaxation? The meta methods have a lot of overlap if not redundant to each other, thus reducing their number and adding a couple of the classical methods would have been more meaningful.

·        What is the added value of having 20 and 50 dimensions? Would only one dimension be enough for the sake of comparison?  Same question and suggestion for Cases I to 6, I would keep the results for example 1 and 4 and move the rest to the appendix, or keep 3 and 6, and move the others to the appendix.

·        The readability of the paper is hindered by having these long tables in the middle. If you were to keep the other dimensions, I would advise moving tables 7, 8, 13, 14, 16 and 17 to the appendix. For Figure 3, I would keep the results a-c and increase their size, and move d-I to the appendix too.  

·        In the text for example, below table 8, which should be moved before table 6, you can add a bit more discussion and qualitative and quantitative comparison. For example, you may say that the CMFO is x% faster than this method. I would also invite you to consider some basic statistical calculations and figure such as box plots and you may explore other ideas too. 

I think the introduction is the part that needs most of the improvement, especially on how to organise it and how to make it more interesting and easier to go through. 

Author Response

Point 1: The introduction needs a little more structure and flow. I just got trying to understand the purpose of it. Try to be more objective and critical when writing it. Response 1: Many thanks for the suggestions. We have modified and optimized the structure and flow of the introduction. Lines 339-342: Therefore, evolutionary computation techniques are usually chosen to solve such problems, including genetic algorithms (GA), particle swarm optimization (PSO), ant colony optimization (ACO), and others. Lines 351-353: However, only low-dimensional cases have been studied, and experiments and analyses of high-dimensional cases are lacking. Lines 356-357: A new parallel hybrid meta-heuristic method, combining the hybrid topology binary particle swarm optimization algorithm, adaptive differential evolution algorithm, and lambda iterative method, is proposed to solve ED problem, however, no smart demand side management is achieved. Lines 359-362: A short-term load ED problem combines the ED problem with machine learning-based short-term load forecasting (STLF), where a new energy dispatch model is proposed and solved by a new dynamic genetic algorithm, however, experiments were conducted on only three test systems and no other new energy generation methods were introduced. Lines 363-366: Due to the uncertainty of wind power, the wind speed model is considered in the ED problem, which is conducive to exploring a more balanced low-carbon power dispatching strategy for wind power integrated systems, and is investigated by the flower pollination algorithm to solve this class of economic dispatching problems. Lines 371-374: The cuckoo search algorithm (CSA) is applied to ED problems where wind turbines and fuel cells are considered, and a cuckoo search algorithm for microgrid power dispatch problems has also been proposed with some prospective. Lines 374-378: The fuzzy dominance is used to select Pareto optimal front (POF) and the multi-objective fuzzy dominance-based bacterial foraging algorithm is proposed to solve the economic emission dispatch problem, however, it was not compared with other multi-objective algorithms, and it was difficult to reflect the superiority of the improved algorithm. Lines 378-382: A multi-objective optimization algorithm based on time-varying accelerated particle swarm optimization has been introduced to the cogeneration problem with a high convergence rate and solving the uncertainty of energy demand and supply for intermittent renewable energy sources, but the method has only been applied to a 7-unit test system and there is a lack of research on other scale test systems. Lines 388-391: By combining stochastic exploratory search and learning strategies, an improved gray wolf optimization (GWO) algorithm is designed to solve the ED problem in different dimensions and effectively reduces the generation cost compared with other algorithms. Lines 393-395: In order to consider the impact of wind power generation, a novel chaotic quantum genetic algorithm was developed to solve the ED problem of wind power generation with good results. Lines 397-400: An improved competitive group optimization algorithm has been proposed to handle both static and dynamic ED problems and minimize the total fuel cost by determining the intraregional generation and interregional power exchange, and has advantages in terms of solution accuracy and reliability compared to other algorithms. Point 2: I would move the paragraph after table 8 before table 6, it would have been better for understanding. Response 2: Many thanks for the suggestions. For better understanding, we have moved the paragraphs after table 8 to before table 6. The modified paragraphs are in lines 662-690. Point 3: Also, I think the wording of N and Y is a bit confusing can you please be clearer with this? Response 3: Many thanks for the suggestions. We have made the meaning of N and Y clearer. Modified content in lines 557-559. Lines 663-665: Among the three metrics "Sta", "Best", and "Time", N means that the algorithm is worse than CMFO in more than two metrics, and Y means that the algorithm is better than CMFO in more than two metrics. Point 4: In Tables 6, 7 and 8 why haven’t you compared it to MFO without the chaotic part? It would have been interesting to see the behaviour with and without this new addition. This is a critical question to answer. Response 4: Many thanks for the suggestions. We have refined the experimental data of MFO into Table 6, Table A1 in Appendix A, and Table A2 in Appendix A. With this improved content, the experiment becomes more complete, more convincing, and more interesting. Again, thank you for the suggestions you provided. Modified content in lines 695-696. Point 5: It is impressive the number of metaheuristic methods that you are comparing to! However, why have you limited that comparison to this method? Why not compare the classical non-heuristic methods such as quadratic programming, gradient descent, and Lagrange relaxation? The meta methods have a lot of overlap if not redundant to each other, thus reducing their number and adding a couple of the classical methods would have been more meaningful. Response 5: Many thanks for the suggestions. Since this thesis is about using metaheuristic algorithms to solve dynamic economic scheduling problems, only metaheuristic algorithms are chosen for comparison when testing the performance of the Chaos Moth Flame algorithm, which can make this thesis more relevant. Of course, the suggestions made are greatly appreciated. Point 6: What is the added value of having 20 and 50 dimensions? Would only one dimension be enough for the sake of com-parison? Same question and suggestion for Cases I to 6, I would keep the results for example 1 and 4 and move the rest to the appendix, or keep 3 and 6, and move the others to the appendix. Response 6: Many thanks for the suggestions. For metaheuristic algorithms, testing only one dimensional international standard algorithm cannot reflect the superior performance of the improved algorithm, because for some algorithms, the performance of the algorithm decreases significantly when the dimensionality is increased, and it is easy to fall into a local optimum, resulting in a poorer quality of the solution. Therefore, in this paper, three dimensions are chosen to test the comprehensive performance of CMFO. The three dimensions correspond to the low-dimensional problem, the middle-dimensional problem, and the high-dimensional problem, and CMFO can be better used in dynamic economic scheduling problems if it can show better performance in all three dimensions. In addition, the dynamic economic dis-patch problem of different sizes (different dimensions) is studied in this paper, so it is also necessary to test cases with different number of units, which can be more relevant to the actual generation situation. Point 7: The readability of the paper is hindered by having these long tables in the middle. If you were to keep the other dimensions, I would advise moving tables 7, 8, 13, 14, 16 and 17 to the appendix. For Figure 3, I would keep the results a-c and increase their size, and move d-I to the appendix too. Response 7: Many thanks for the suggestions. We are aware of this issue and have moved the section you suggested to the appendix. Furthermore, modified content in lines 980-1055. Point 8: In the text for example, below table 8, which should be moved before table 6, you can add a bit more discussion and qualitative and quantitative comparison. For example, you may say that the CMFO is x% faster than this method. I would also invite you to consider some basic statistical calculations and figure such as box plots and you may explore other ideas too. Response 8: Many thanks for the suggestions. We have refined the content and added a comparison of quantitative and qualitative analysis. Modified content in lines 671-682. Lines 671-682: For example, in 10 dimensions, CMFO converged 305%, 916%, 1511%, 135%, 2%, 163%, 24%, 147%, 77%, and 9% faster compared to the other 10 algorithms in the unimodal test function F1, and also improved in the stability of the optimal values. In the multimodal test function F6, CMFO improves the convergence speed compared to the other 10 algorithms by 1544%, 314%, 589%, 52%, -3%, 104%, -1%, 110%, 18%, and 7%, respectively, and improves the stability and accuracy of the optimal values. In the multimodal test function F7, CMFO converges 102%, 339%, 684%, 62%, 9%, 121%, 8%, 121%, 38%, and 4% faster than the other 10 algorithms, respectively, and also improves in the accuracy of the optimal values. Moreover, the advantages of CMFO in optimizing both unimodal and multimodal test functions remain when raised to 20 and 50 dimensions, indicating that the improved algorithm is equally suitable for solving high-dimensional, high-complexity optimization problems.

Reviewer 2 Report

The article is devoted to the issues of dynamic economic optimization, the solution of which makes it possible to increase the efficiency of energy units and therefore belongs to the relevant ones.

Remarks.

1. It is unclear why the authors limit a number of types of power plants involved in the generation of power and energy (p.2.1.).

2. It is also unclear why the power ratio of generators of stations and electric vehicles is not given? How great is the influence of the latter? Is this confirmed?

3. The authors have not sufficiently justified the use of optimization methods.

4. The article is overloaded with a table of calculation results, they should be reduced

5. The conclusion talks about carbon neutrality as a justification for the relevance of the study, but at the same time it is stated about the prospects of renewable sources of power and energy that are carbon-neutral. Conclusions should be formulated correctly.

Author Response

Point 1: It is unclear why the authors limit a number of types of power plants involved in the generation of power and energy (p.2.1.). Response 1: Many thanks for the suggestions. In this paper, we study two types of power generation from thermal power units and plug-in electric vehicles, therefore, we do not deal with wind power or solar power as types of power plants. We will consider including various types of power plants in future articles, and your suggestions are greatly appreciated. Point 2: It is also unclear why the power ratio of generators of stations and electric vehicles is not given? How great is the influ-ence of the latter? Is this confirmed? Response 2: Many thanks for the suggestions. We have added this aspect, and the modifications are in lines 765-777. Lines 765-777: In addition, the charging and discharging power of PEVs is 100MW, and for 5 units, the maximum pow-er generation of a single unit is 300MW, and the minimum value of power generation demand for each time period is 410MW and the maximum value is 720MW. For 10 units, the maximum generation power of each single unit is 470MW, and the minimum value of generation demand for each time period is 1036MW and the maximum value is 1972MW. For 15 units, the maximum generation power of each single unit is 470 MW, and the minimum value of generation de-mand for each time period is 1171 MW and the maximum value is 2394 MW. Therefore, from the power generation perspective, the power ratios of a single genset and an PEVs are 3:1, 4.7:1 and 4.7:1, respectively, and the impact of an PEVs is 13.9%~ 24.4% for 5 units, 5.1%~9.7% for 10 units and 4.2%~8.5% for 15 units. Therefore, the reasonable deploy-ment of electric vehicle charging and discharging can reduce the grid fluctuation to a certain extent and achieve the ef-fect of peak and valley reduction. Point 3: The authors have not sufficiently justified the use of optimization methods. Response 3: Many thanks for the suggestions. We have refined the rationale for using the optimization method, and the modifica-tions are in lines 668-683. Lines 668-683: In particular, the small optimal value and standard deviation of different kinds of functions indicate that CMFO has high solution precision and stability. Furthermore, compared with other methods, CMFO significantly re-duces the time of computing. For example, in 10 dimensions, CMFO converged 305%, 916%, 1511%, 135%, 2%, 163%, 24%, 147%, 77%, and 9% faster compared to the other 10 algorithms in the unimodal test function F1, and also improved in the stability of the optimal values. In the multimodal test function F6, CMFO improves the convergence speed com-pared to the other 10 algorithms by 1544%, 314%, 589%, 52%, -3%, 104%, -1%, 110%, 18%, and 7%, respectively, and im-proves the stability and accuracy of the optimal values. In the multimodal test function F7, CMFO converges 102%, 339%, 684%, 62%, 9%, 121%, 8%, 121%, 38%, and 4% faster than the other 10 algorithms, respectively, and also improves in the accuracy of the optimal values. Moreover, the advantages of CMFO in optimizing unimodal and multimodal test func-tions remain when improved to 20 and 50 dimensions, indicating that the improved algorithm is equally suitable for solving high-dimensional, high-complexity optimization problems, making it a good choice to use CMFO for problems with higher dimensionality and complexity like DED. Point 4: The article is overloaded with a table of calculation results, they should be reduced Response 4: Many thanks for the suggestions. We are aware of this problem and have reduced the unnecessary tables of calculated results. After consideration, we have removed Table 12, Table 13, and Table 14 from the original paper, which makes the article more concise. Point 5: The conclusion talks about carbon neutrality as a justification for the relevance of the study, but at the same time it is stated about the prospects of renewable sources of power and energy that are carbon-neutral. Conclusions should be formulated correctly. Response 5: Many thanks for the suggestions. We have revised the conclusions. Modified content in lines 928-940. Lines 928-940: This paper explores a modified version of the Moth Flame Algorithm (MFO), referred to as CMFO, to solve DED problem in the presence of PEVs in a more efficient manner. The proposed algorithm incorporates chaotic map-ping to initialize the population and improve the evolution mechanism of MFO, thereby compensating for its short-comings and avoiding premature convergence. The performance of CMFO is evaluated through experiments on bench-mark functions and compared with other popular algorithms. The results demonstrate that CMFO can find high-quality solutions in a shorter time frame. Furthermore, two scenarios with and without PEVs are tested using CMFO, with the number of units ranging from 5 to 15. The results show that CMFO can produce optimal solutions, especially for large-scale problems and confirm that V2G can effectively reduce grid fluctuations and achieve peak shaving and valley filling. In the future, we will consider the introduction of solar energy as well as wind energy into the DED problem, which will be more helpful to solve the actual problem.

Reviewer 3 Report

Dear Authors,

The paper proposed a model that integrates dynamic economic dispatch (DED) with plug-in electric vehicles (PEVs) to mitigate grid fluctuations and balance peaks and valleys. An improved chaos moth flame optimization algorithm (CMFO) is introduced to solve the model, which has better global optimization capabilities.

The feasibility of the proposed model is validated through numerical experiments on benchmark functions and various generation units of different sizes using the improved chaos moth flame optimization algorithm (CMFO).

The numerous case studies presented to demonstrate the advantages of CMFO over the existing metaheuristics is the major contribution of the paper. However, the paper has scope for further improvement.

 Please discuss the difference between economic dispatch (ED) and dynamic economic dispatch (DED).

 The coordination between the economic dispatch of power plants and the charging/discharging of numerous electric vehicles (EVs) must be explained in detail.

 Please discuss how the optimization formulation will change with the incorporation of PV and Wind power plants.

 The proposed idea to mitigate grid fluctuations using PEVs can also be achieved using large scale energy storage. In this context, please discuss the advantage of using PEVs instead of energy storage.

 Please explain figures 6 and 7 in detail. The existing discussion is not sufficient

If possible implement the case studies on an IEEE test system.

Please discuss the simulation program used to solve the optimization algorithm.

English language is fine.

Author Response

Point 1: Please discuss the difference between economic dispatch (ED) and dynamic economic dispatch (DED). Response 1: Many thanks for the suggestions. We have revised the paper. Modified content in lines 409-414. Lines 409-414: Economic dispatch includes static economic dispatch and dynamic economic dispatch. Static economic dispatch assumes constant load, which is difficult to adapt to the dynamic changes of actual power load. Dynamic economic dispatch generally divides a day into 24 hours and optimizes daily economic dispatch according to daily load forecast, which is more in line with the actual power system operation. The mainstream of current research is dynamic economic dispatch. Point 2: The coordination between the economic dispatch of power plants and the charging/discharging of numerous electric vehicles (EVs) must be explained in detail. Response 2: Many thanks for the suggestions. We have revised the paper. Modified content in lines 417-434. Lines 417-434: For large-scale PEVs, a distributed access and central management approach is generally adopted. The PEVs dispatching center can exchange energy and information directly with the power grid, and then the PEVs dispatching center directs the energy exchange of each vehicle according to the actual situation of the serviceable PEVs, so that orderly charging and discharging coordinated management can be realized. In addition, the conversion speed of PEVs charging and discharging is very fast, so it can be seen as a distributed energy storage device that can be charged and discharged, and the real-time power flow between PEVs and grid can be realized through bi-directional power con-version technology, which is also known as electric vehicle to grid (V2G). By accurately transmitting information in both directions between the dispatch center and the PEVs, it is possible to achieve bi-directional, real-time and controllable energy conversion. Of course, this involves the integration of various technologies such as power electronics, communication, power scheduling, and load forecasting. Therefore, it is possible to control the PEVs for orderly discharging when in the peak period of electricity consumption, and control the PEVs for orderly charging when in the low peak period of electricity consumption, so as to reduce the burden of the power grid to a certain extent, improve the safety of the power grid, and achieve the purpose of coordinated charging and discharging. Point 3: Please discuss how the optimization formulation will change with the incorporation of PV and Wind power plants. Response 3: Many thanks for the suggestions. This article has not yet considered how to deploy PV and wind power plants after adding them, we will take this into account in future articles, hope you can understand, thank you very much for your suggestions. Point 4: The proposed idea to mitigate grid fluctuations using PEVs can also be achieved using large scale energy storage. In this context, please discuss the advantage of using PEVs instead of energy storage. Response 4: Many thanks for the suggestions. We have improved the content. Modified content in lines 435-439. Lines 435-439: Compared with the ordinary large-scale energy storage, the use of electric vehicles as a buffer for the grid is due to the fact that electricity can flow in both directions between the grid and PEVs. Moreover, as the number of PEVs increases, it is necessary to rationalize their management, otherwise the grid will be overloaded when the number of PEVs reaches a certain scale. Point 5: Please explain figures 6 and 7 in detail. The existing discussion is not sufficient Response 5: Many thanks for the suggestions. We have explained Figure 6 and Figure 7. Modified content in lines 794-800. Lines 794-800: In Figure 6, the "FEVs" curve represents the charge and discharge of PEVs in each period, and the "Origi-nal Demand + FEVs" curve represents the actual power generation of the genset in each period when PEVs are engaged. The curve "Original Demand + FEVs" reduces the grid fluctuation compared with the curve "Original Demand" and achieves the effect of peak and valley reduction. Each stacked histogram in Figure 7 represents the sum of the power generation of each test case at each period, and the bars with different colors represent the power output of different units. Point 6: If possible implement the case studies on an IEEE test system. Response 6: Thank you very much for your suggestion. We are using the test data of IEEE 5 unit, IEEE 10 unit and IEEE 15 unit, but in order to avoid the length of the article, we do not show the relevant unit parameters in the article, I hope you can forgive us and thank you again for your suggestion. Point 7: Please discuss the simulation program used to solve the optimization algorithm. Response 7: Many thanks for the suggestions. We have given detailed pseudocode in the paper for the reader to read and understand, and the relevant programs are in Table 1 of the article. In order to avoid the length of the article, we won't go into details of the simulation procedure here. We hope you will forgive us and thank you again for your suggestions.

Round 2

Reviewer 1 Report

Thanks for addressing the comments and feedback. Best of luck. 

Moderate editing maybe needed. 

Reviewer 3 Report

Dear Authors,

Thank you for answering all my queries. 

Good luck with all your projects.